# Comments on one-form global symmetries and their gauging in 3d and 4d

Po-Shen Hsin[1,2*], Ho Tat Lam[1†] and Nathan Seiberg[3‡]

**1** Physics Department, Princeton University, Princeton, NJ, USA
**2** Walter Burke Institute for Theoretical Physics,
California Institute of Technology, Pasadena, CA 91125, USA
**3** School of Natural Sciences, Institute for Advanced Study, Princeton, NJ, USA

★ phsin@caltech.edu, † htlam@princeton.edu, ‡ seiberg@ias.edu

## Abstract

We study 3d and 4d systems with a one-form global symmetry, explore their consequences, and analyze their gauging. For simplicity, we focus on $\mathbb{Z}_N$ one-form symmetries. A 3d topological quantum field theory (TQFT) $\mathcal{T}$ with such a symmetry has $N$ special lines that generate it. The braiding of these lines and their spins are characterized by a single integer $p$ modulo $2N$. Surprisingly, if $\gcd(N,p) = 1$ the TQFT factorizes $\mathcal{T} = \mathcal{T}' \otimes \mathcal{A}^{N,p}$. Here $\mathcal{T}'$ is a decoupled TQFT, whose lines are neutral under the global symmetry and $\mathcal{A}^{N,p}$ is a minimal TQFT with the $\mathbb{Z}_N$ one-form symmetry of label $p$. The parameter $p$ labels the obstruction to gauging the $\mathbb{Z}_N$ one-form symmetry; i.e. it characterizes the 't Hooft anomaly of the global symmetry. When $p = 0 \bmod 2N$, the symmetry can be gauged. Otherwise, it cannot be gauged unless we couple the system to a 4d bulk with gauge fields extended to the bulk. This understanding allows us to consider $SU(N)$ and $PSU(N)$ 4d gauge theories. Their dynamics is gapped and it is associated with confinement and oblique confinement – probe quarks are confined. In the $PSU(N)$ theory the low-energy theory can include a discrete gauge theory. We will study the behavior of the theory with a space-dependent $\theta$-parameter, which leads to interfaces. Typically, the theory on the interface is not confining. Furthermore, the liberated probe quarks are anyons on the interface. The $PSU(N)$ theory is obtained by gauging the $\mathbb{Z}_N$ one-form symmetry of the $SU(N)$ theory. Our understanding of the symmetries in 3d TQFTs allows us to describe the interface in the $PSU(N)$ theory.

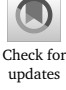
# 1  Introduction and Summary

## 1.1  One-form symmetries

Point operators can be charged under an ordinary internal global symmetry. Extended operators can be charged under a higher-form global symmetry [1]. One-form symmetries characterize line operators, two-form symmetries characterize surface operators, etc. One of the points of [1] is that many of the standard properties of ordinary global symmetries are present also in the case of their higher-form generalizations.

- The symmetries might or might not be spontaneously broken. If they are unbroken, the spectrum includes charged states. For example, when a one-form global symmetry is unbroken the spectrum includes charged strings. If they are broken, the low-energy dynamics reflects the broken symmetry. For example, if the global symmetry is discrete and the spectrum is gapped, the low-energy theory includes a TQFT.

- As with ordinary symmetries, higher-form symmetries can have 't Hooft anomalies. Such anomalies obstruct their gauging. These anomalies can be used, just like 't Hooft anomaly matching of ordinary global symmetries, to constrain the IR behavior of a theory and to check duality between distinct theories. Also, such an anomaly in a higher-form symmetry can flow from a bulk to a defect in the bulk.

Unlike ordinary global symmetries, higher-form symmetries must be Abelian. In this note we will focus mostly on $\mathbb{Z}_N$ one-form global symmetries in 3 and 4 dimensions. Typical examples in 3d are $U(1)_N$ or $SU(N)_k$ Chern-Simons (CS) theory. They have a spontaneously broken $\mathbb{Z}_N$ one-form symmetry.

A typical example in 4d is an $SU(N)$ gauge theory without quarks. Here the $\mathbb{Z}_N$ one-form symmetry is expected to be unbroken, which is related to the confinement of the system. If we add quarks in the fundamental representation to this theory, then the one-form symmetry is absent, and indeed the theory with quarks does not have a meaningful notion of confinement.

## 1.2  4d $SU(N)$ gauge theory with $\theta$ and domain walls

Of particular interest for us will be the behavior of this 4d $SU(N)$ theory with a $\theta$-parameter. The lore is that at generic $\theta$ the system is confining and gapped with a trivial vacuum. At $\theta \in \pi\mathbb{Z}$, we have time-reversal and parity symmetries. These are unbroken at $\theta \in 2\pi\mathbb{Z}$. (For small values of $N$ there are also other logical options [2].) But they are spontaneously broken at $\theta$ an odd multiple of $\pi$. In these cases the system has two degenerate vacua with domain walls that interpolate between them. Arguments based on anomalies in the one-form symmetry, which we will review below, suggest that the theory on the domain wall is an $SU(N)_1$ TQFT [1, 2].[1]

As stressed in [1, 2], the transition at $\theta = \pi$ separates two distinct vacua in the following sense. On one side of the transition monopoles condense, leading to confinement, and on the other side of the transition dyons condense, leading to oblique confinement. More precisely, the transition at $\theta$ an odd multiple of $\pi$ separates two distinct oblique confinement vacua. Since different dyons condense on the two sides of the domain wall, no dyon condenses on the wall. Therefore, the theory on the wall is not confining and the Wilson lines of the $SU(N)_1$ theory on the wall are world lines of unconfined probed quarks. Not only are these quarks liberated, they also have nontrivial braiding, i.e. they are anyons! Below we will give an intuitive physical argument explaining why they are anyons.

## 1.3  Interfaces

One of our goals is to study in detail interfaces in this theory. We let $\theta$ be a space- dependent interpolation between $\theta_0$ to $\theta_0 + 2\pi k$. If the interpolation is over a length scale much longer than the inverse of the dynamical scale of the theory $\Lambda$, then at a generic spacetime point $\theta$ is essentially constant on the scale where confinement takes place and the vacuum is unique and varies smoothly. When $\theta$ crosses an odd multiple of $\pi$ there is a domain wall separating two vacua. Therefore, the interpolation leads to $k$ domain walls with $SU(N)_1$ on each of them [2], as illustrated in Figure 1a. If the interpolation is more rapid, then the TQFT $SU(N)_1 \otimes SU(N)_1 \otimes \dots$ can undergo a transition to another TQFT $\mathcal{T}_k$, see Figure 1b. It was suggested in [2, 3] that this theory is $SU(N)_k$. However, we will soon argue that there are also other logical possibilities and only a more detailed dynamical analysis can determine the right answer.

It is important that the theory on the interface is uniquely determined by the microscopic theory and by the profile of the space-dependent $\theta$. This is to be contrasted with a sharp

---

[1]Although as spin TQFTs $SU(N)_1 \longleftrightarrow U(1)_{-N}$, we prefer to use $SU(N)_1$ because our theory is bosonic.

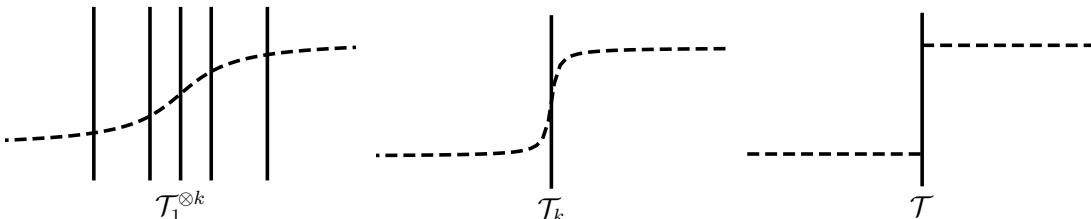

(a) Slow $\theta$ interpolation     (b) Rapid $\theta$ interpolation     (c) Sharp $\theta$ interpolation

Figure 1: The interfaces for different profiles of $\theta$ that interpolate from $\theta = \theta_0$ to $\theta = \theta_0 + 2\pi k$. The dashed lines are the profile of the $\theta$ parameter and the solid lines are the locations of the interfaces. In (a), there are $k$ domain walls located at the transitions when $\theta$ crosses an odd multiple of $\pi$. The theory on each domain wall is $\mathcal{T}_1$, which we argue is $SU(N)_1$ [1]. When the $\theta$ variation is more rapid, as in (b), there is only one interface and the theory on it is $\mathcal{T}_k$. One option for that theory is $SU(N)_k$, but we will argue that other options are also possible. Finally, as in (c), $\theta$ can be discontinuous. In this case the theory on the interface $\mathcal{T}$ is not determined uniquely by the microscopic dynamics. But it is constrained by anomaly considerations.

interface when $\theta$ is discontinuous, as illustrated in Figure 1c. Here we have the freedom to change the theory on the interface by adding more degrees of freedom there and to consider their dynamics. We will not study it here. The same comments apply to a system with a boundary. As with the sharp interface, the boundary theory is constrained by anomalies, but there is a lot of freedom in adding boundary degrees of freedom.

Our main tool for analyzing the system is its $\mathbb{Z}_N$ one-form global symmetry. Related to this symmetry is an integer label $p$ with $p \sim p + 2N$ and $pN$ even [1,4]. Furthermore, we have an identification in labeling the theories [1,2,4]

$$(\theta, p) \sim (\theta + 2\pi k, p + k(N-1)) . \tag{1.1}$$

One way to think about the parameter $p$ is through coupling the $\mathbb{Z}_N$ global symmetry to a classical background two-form gauge field $\mathcal{B}_\mathcal{C}$ (the subscript $\mathcal{C}$ means that it is classical). Then, the parameter $p$ is the coefficient of a counterterm proportional to the square of $\mathcal{B}_\mathcal{C}$ [1,4]. This term does not affect any separated points correlation function, but it does affect contact terms and the behavior of the system with a boundary.

The key dynamical fact is that the theory confines. This means that the $\mathbb{Z}_N$ one-form symmetry is unbroken. Also, the spectrum is gapped and the low-energy dynamics is trivial – there is not even a TQFT at long distances. The only meaningful fact that remains at low energies is the coefficient $p$ of the counterterm of $\mathcal{B}_\mathcal{C}$, which means that the system can be in a nontrivial Symmetry Protected Topological (SPT) phase.

When we have an interface where $\theta$ changes by $2\pi k$ the two sides of the interface are typically in *different* SPT phases labeled by $p^\pm$ with

$$p^+ - p^- = k(N-1) \bmod 2N . \tag{1.2}$$

This means that when $p^+ \neq p^- \bmod 2N$ the theory on the interface cannot be trivial. It must have a $\mathbb{Z}_N$ one-form global symmetry with anomaly $(p^+ - p^-) \bmod 2N$.

Let us try to determine the theory on the interface. When the interface is rapid, we can shift $\theta$ on one side, as in equation (1.2), so that $\theta$ does not change across the interface, but

$p$ changes. It induces a Chern-Simons term $SU(N)_k$ on the interface. Next, as the theory becomes strongly coupled it confines and the bulk on the two sides of the interface become gapped and trivial. What happens to the $SU(N)_k$ theory on the interface? One option, which was advocated in [2], is that at least for small enough $|k|$ it is not affected by the confinement. However, the strong dynamics could change that answer.[2] But whatever the dynamics does, the one-form $\mathbb{Z}_N$ global symmetry and its anomaly $p^+ - p^-$ cannot change. Therefore, if $p^+ \neq p^-$ mod $2N$, the theory on the interface cannot be trivial, and we'll denote it by $\mathcal{T}_k$.

We start by reconsidering the special case $k = 1$. Can the UV answer $SU(N)_1$ be modified? We suggest that this cannot happen. First, as we will discuss in detail below, this particular theory is the minimal theory with a $\mathbb{Z}_N$ one-form symmetry of anomaly $N - 1$. Every other TQFT with this property factorizes into $SU(N)_1$ times another TQFT, whose line operators are $\mathbb{Z}_N$ invariant. Therefore, it is natural to assume that in this case the UV answer does not change. Also, in a closely related supersymmetric theory, a string construction shows that the theory on the interface is $U(1)_{-N}$ [5], which is dual (as spin TQFT) to our answer $SU(N)_1$ [6].

As we move to higher values of $k$ the situation is less clear. It was suggested in [2] that as a slow interface becomes steeper, the $SU(N)_1^{\otimes k}$ TQFT can be Higgsed to the diagonal $SU(N)_k$. This would agree with the answer in the UV. However, further dynamical effects can change this answer. Since we expect the interface theory to remain non-confining, we do not anticipate monopoles to participate in this dynamics on the interface. Instead, we can consider dynamical scalar fields in the adjoint representation of $SU(N)$. Such scalar fields can arise from modes of the microscopic gluons and their presence does not break the exact $\mathbb{Z}_N$ one-form symmetry of the system. The condensation of these scalars can Higgs $SU(N)$ to various subgroups. The maximum possible Higgsing with one adjoint scalar is to the Cartan torus $U(1)^{N-1}$. In this case the $SU(N)_k$ theory becomes $U(1)^{N-1}$ with a coefficient matrix given by $kK_{\text{Cartan}}$ with $K_{\text{Cartan}}$ the Cartan matrix of $SU(N)$. (Note that for $k = 1$ the TQFT $SU(N)_1$ is the same as this Abelian TQFT.) With more than one adjoint scalars, we can further Higgs the system all the way down to a $\mathbb{Z}_N$ gauge theory[3] with level $K = -kN(N - 1) = -(p^+ - p^-)N$. Below we will review in detail this TQFT and its properties.

The upshot of the discussion above is that the spontaneously broken $\mathbb{Z}_N$ one-form symmetry and its anomaly $p^+ - p^-$ restrict the TQFT on the interface $\mathcal{T}_k$, but do not uniquely determine it. For $k = 1$ it is natural to assume that the correct answer is the minimal one $\mathcal{T}_1 = SU(N)_1$. For higher values of $k$ there are several natural possibilities including $SU(N)_k$, but the other options include also some Abelian TQFTs. It should be emphasized, however, that despite our inability to determine $\mathcal{T}_k$ beyond the symmetry and anomaly constraints, this theory is uniquely determined by the dynamics.

## 1.4 Gauging the $\mathbb{Z}_N$ one-form symmetry – 4d $PSU(N)$ gauge theory and interfaces

When the $\mathbb{Z}_N$ one-form symmetry is gauged, the microscopic 4d $SU(N)$ gauge theory becomes a $PSU(N)$ gauge theory and the macroscopic theory might no longer remain trivial [10]. Specifically, it becomes a $\mathbb{Z}_L$ gauge theory with[4]

$$L = \gcd(p, N) . \tag{1.4}$$

---

[2]We thank E. Witten for encouraging us to think about other options.

[3]The $\mathbb{Z}_N$ gauge theory at level $K$ can be expressed as the following $U(1) \times U(1)$ Chern-Simons theory [4,7,8]

$$(\mathcal{Z}_N)_K : \quad \int \left( \frac{K}{4\pi} x dx + \frac{N}{2\pi} x dy \right) . \tag{1.3}$$

For even $K$ this is a Dijkgraaf-Witten (DW) theory [9].

[4]Below we will show that on a nonspin manifold this $\mathbb{Z}_L$ gauge theory is sometimes twisted in a particular way.

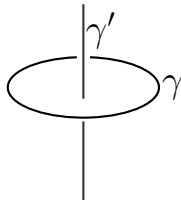

Figure 2: Braiding the line operators supported on the curves $\gamma$ and $\gamma'$.

Unlike the original $SU(N)$ theory where $p$ affects only the SPT phase, here it affects the low-energy dynamics. Now the interface is more interesting. Clearly, we have a $\mathbb{Z}_{L_\pm}$ gauge theory with $L_\pm = \gcd(p_\pm, N)$ on the two sides of the interface. But what is the resulting theory on the interface?[5]

When $L_+ = L_- = 1$ the bulk theory on the two sides is trivial and the low-energy theory is only the 3d theory on the interface and it is completely meaningful. However, when either $L_+$ or $L_-$ (or both) are not equal to one, the bulk theory is not trivial and the low-energy TQFT is not three dimensional. It is four dimensional and the interface appears as a 3d defect in the 4d bulk. Therefore, it is meaningless to ask what the 3d theory on the interface is. It is not decoupled from the 4d bulk. Nevertheless, we will argue that there exists a 3d TQFT that captures many of the features of the physics along the interface. Roughly, it is a quotient of the full 4d system by the physics of the 4d bulk. We will describe this in more detail below.

## 1.5 One-form global symmetries in 3d and their gauging

In order to understand these TQFTs we will have to explore in more detail the one-form global symmetry, its anomaly, and its gauging in 3 and 4 dimensions. Let us start with a 3d one-form symmetry $\mathcal{A}$. The charge operators are line operators $a_{\mathbf{g}}$ labeled by a group element $\mathbf{g} \in \mathcal{A}$. The group multiplication corresponds to the fusion of the lines:

$$a_{\mathbf{g}+\mathbf{g}'} = a_{\mathbf{g}} a_{\mathbf{g}'} \, , \tag{1.5}$$

where the group multiplication of $\mathcal{A}$ is denoted by addition, and the product of two lines denotes their fusion. Each line $a_{\mathbf{g}}$ represents an Abelian anyon in the TQFT.

For simplicity we will focus on a $\mathbb{Z}_N$ one-form symmetry. The symmetry lines are $a^s$ with

$$a^N = 1 \tag{1.6}$$

and we refer to $a$ as the generating line. In general, this generator is not unique and some of the expressions below depend on the choice of generator.

In a TQFT with a $\mathbb{Z}_N$ one-form symmetry, each line $W$ carries a $\mathbb{Z}_N$ charge $q(W) \in \mathbb{Z}_N$ under the symmetry, which is determined by braiding the generating line $a$ with $W$ (see Figure 2):

$$a(\gamma)W(\gamma') = W(\gamma')e^{\frac{2\pi i q(W)}{N}} \, . \tag{1.7}$$

We will show that general considerations constrain the spins of the symmetry lines to be[6]

$$h[a^s] = \frac{ps^2}{2N} \bmod 1 \, , \tag{1.8}$$

---

[5]Note that the naive answer $PSU(N)_k$ cannot be right. For generic $k$ this is not a consistent theory [9, 11]!

[6]We thank Z. Komargodski and J. Gomis for a discussion about this point.

for some integer $p = 0, 1, \cdots, 2N - 1 \mod 2N$. Imposing (1.6) leads to

$$pN \in 2\mathbb{Z} \,. \tag{1.9}$$

The situation in spin TQFT is slightly different because such theories have a transparent spin-half line $\psi$. This will be discussed in detail below.

One significance of the parameter $p$ is that it determines the $\mathbb{Z}_N$ charge $q(a) = -p \mod N$ of the generating line $a$ (see Section 2.1). Clearly, the symmetry can be gauged only when the symmetry lines themselves are neutral, i.e. when $q(a) = 0$. Therefore, the parameter $p$ controls the obstruction to gauging, which is the 't Hooft anomaly.

When $p = 0$, the $\mathbb{Z}_N$ one-form symmetry is anomaly free and it can be gauged. Denoting the original TQFT by $\mathcal{T}$, we will denote the result of this gauging by the TQFT

$$\mathcal{T}' = \mathcal{T}/\mathbb{Z}_N \,. \tag{1.10}$$

When $p = N$ the generating line has spin $\frac{1}{2}$ and the gauged system $\mathcal{T}/\mathbb{Z}_N$ is a spin TQFT.[7]

There are several ways to describe the gauging procedure. From the perspective of symmetry defects, gauging a symmetry amounts to summing over all possible insertions of symmetry defects [1]. In the corresponding two-dimensional chiral algebra, gauging the one-form symmetry corresponds to extending the chiral algebra [11, 12]. For Chern-Simons theory it can sometimes be described by the quotient of the gauge group by a subgroup of the center [1, 11]. In the condensed matter literature, it is called "anyon condensation" of the Abelian anyon that corresponds to the generating line of the one-form symmetry [13].

For $p = 0$ when the symmetry generating line $a$ has integer spin the gauging involves three steps [11, 12]:

Step 1  Discard the lines $W$ that are not invariant under the $\mathbb{Z}_N$ one-form symmetry.

Step 2  Since $a$ is trivial, we identify the lines $W$ and $Wa$ obtained by fusing with $a$.

Step 3  If $W$ is a fixed point under the fusion with $a$, then there are $N$ copies of $W$. More precisely, if $s$ is the minimal divisor of $N$ such that $W$ is invariant under the fusion with $a^s$, then there are $N/s$ copies of $W$.[8]

For even $p = N$, the generating line $a$ has half-integer spin and then the resulting theory after gauging is a spin TQFT. As we will discuss below, this leads to the same three-step process.

When $p \neq 0, N$ the generating line $a$ is charged under the $\mathbb{Z}_N$ symmetry and that symmetry cannot be gauged. However, a subgroup $\mathbb{Z}_L \subset \mathbb{Z}_N$ with[9]

$$L = \gcd(p, N) \tag{1.11}$$

can be gauged. It is generated by the line $\widehat{a} = a^{N/L}$. Since its spin is $h = \frac{pN}{2L^2}$, its $p$-parameter is $\widehat{p} = \frac{pN}{L} \mod 2L$. Note that $\widehat{p} = 0 \mod L$. When $\widehat{p} = 0 \mod 2L$ we can gauge this $\mathbb{Z}_L$ subgroup as above, and when $\widehat{p} = L \mod 2L$ the resulting gauged theory is a spin TQFT. The most anomalous case has $L = 1$ and it will have particular significance below.

---

[7]This is the case even when the original TQFT is non-spin. In this case we can say that there is a mixed 't Hooft anomaly between the $\mathbb{Z}_N$ one-form symmetry and gravity (the bosonic Lorentz symmetry).

[8]This can be proven by iteration. Let $N_1$ be the highest non-trivial divisor of $N_0 = N$. Then gauging the $\mathbb{Z}_{N_0/N_1}$ subgroup generated by $a^{N_1}$ leads to $N_0/N_1$ copies at each fixed point. We can continue to gauge the remaining $\mathbb{Z}_{N_1}$ symmetry by repeating the process. For the minimal divisor $N_i$ such that $W$ is the fixed point under the fusion with $a^{N_i}$, there will be $\frac{N_0}{N_1}\frac{N_1}{N_2} \cdots \frac{N_{i+1}}{N_i} = \frac{N}{N_i}$ copies of $W$ after gauging the $\mathbb{Z}_N$ symmetry.

[9]The relation to the seemingly unrelated equation (1.4) will be clear soon.

## 1.6 Outline and summary of new results

In Section 2 we will discuss in detail the one-form symmetry in 3d and will prove the statements above. We will also show that for given relatively prime $N$ and $p$ (i.e. $L = 1$) there is a minimal TQFT with a $\mathbb{Z}_N$ one-form symmetry of anomaly $p$. We will denote it by $\mathcal{A}^{N,p}$. Furthermore, we will show that any TQFT $\mathcal{T}$ with such a one-form global symmetry factorizes as

$$\mathcal{T}' \otimes \mathcal{A}^{N,p} \qquad \text{for} \qquad L = \gcd(N,p) = 1 \ . \tag{1.12}$$

This means that all the lines in $\mathcal{T}'$ are $\mathbb{Z}_N$ neutral. This is quite surprising – the entire effect of the global symmetry is limited to this factor of $\mathcal{A}^{N,p}$ and the rest of the theory is not affected by it. We can also invert equation (1.12) and map the TQFT $\mathcal{T}$ to

$$\mathcal{T}' = \frac{\mathcal{T} \otimes \mathcal{A}^{N,-p}}{\mathbb{Z}_N} \ . \tag{1.13}$$

When $L = N$ we have the three-step gauging procedure we discussed above that maps a TQFT $\mathcal{T}$ to $\mathcal{T}' = \mathcal{T}/\mathbb{Z}_N$ (1.10). In the other extreme of $L = 1$ we can map $\mathcal{T}$ to $\mathcal{T}'$ of (1.13). Here we simply remove the non-invariant lines, i.e. we perform only step 1 of the three steps.

In Section 2.5 we will generalize this procedure to generic $L = \gcd(N,p)$. We map

$$\mathcal{T} \to \mathcal{T}' = \frac{\mathcal{T} \otimes \mathcal{A}^{N/L,-p/L}}{\mathbb{Z}_N} = \frac{\mathcal{T}/\mathbb{Z}_L \otimes \mathcal{A}^{N/L,-p/L}}{\mathbb{Z}_{N/L}} \ . \tag{1.14}$$

The equality between these expressions will be derived in Section 2. In the map (1.14) we perform step 1 of the three-steps using $\mathbb{Z}_N$ and perform steps 2 and 3 using $\mathbb{Z}_L$. This expression coincides with (1.10) for $L = N$ and with (1.13) for $L = 1$ and generalizes them to generic $L$. (Depending on the details (1.14) might be a spin TQFT.)

This generalized gauging procedure has a physical interpretation, which we describe below, in terms of coupling the system to a 4d bulk gauge theory. It is also related to a more mathematical discussion in [14–17] and the discussion on the Walker-Wang lattice models in [18, 19].

In Section 3, we couple the 3d system to a 4d bulk and promote the background $\mathcal{B}_\mathcal{C}$ gauge fields to quantum fluctuating fields and correspondingly, we drop the subscript $\mathcal{C}$. The bulk theory becomes effectively a $\mathbb{Z}_L$ gauge theory.

As we said above, for $L = 1$ the bulk theory is trivial and therefore there is a meaningful 3d TQFT on the boundary. It cannot be $\mathcal{T}/\mathbb{Z}_N$ because the anomaly makes this quotient inconsistent. Instead, we will show that the theory on the boundary is $\mathcal{T}'$ of (1.13)

$$\mathcal{T}' = \frac{\mathcal{T} \otimes \mathcal{A}^{N,-p}}{\mathbb{Z}_N} \ . \tag{1.15}$$

This equation has several complementary interpretations. First, we can say that the bulk produces a factor of our minimal theory $\mathcal{A}^{N,-p}$ on the boundary such that the combined boundary theory $\mathcal{T} \times \mathcal{A}^{N,-p}$ is anomaly free and then we can gauge the $\mathbb{Z}_N$ symmetry using the three steps above. Second, $\mathcal{T}'$ is as in (1.12), i.e. it includes only the $\mathbb{Z}_N$ invariant lines in $\mathcal{T}$. This means that it is obtained from $\mathcal{T}$ by applying only step 1 of the three-step gauging procedure above. And since $L = 1$ this leads to a consistent TQFT.

When $L \neq 1$ it is not meaningful to discuss the boundary theory, because it does not decouple from the bulk, which includes a non-trivial 4d TQFT. We could attempt to consider a 3d theory that consists only of the lines on the boundary and describes their correlation functions. We will find that these lines are the $\mathbb{Z}_N$ invariant lines from $\mathcal{T}$. This amounts to implementing step 1 of the three-step gauging procedure above. Because of the lack of decoupling from the

bulk, the resulting theory is not a consistent 3d TQFT. It includes $L$ lines that can move from the boundary to the 4d bulk and therefore they have trivial braiding with every line on the boundary. It is natural to consider a new effective theory obtained by performing a quotient by these lines.[10] In more detail, we performed step 1 of the three-step procedure above for $\mathbb{Z}_N$, and now we perform steps 2 and 3 with respect to the $\mathbb{Z}_L$ subgroup. The resulting TQFT is $\mathcal{T}'$ of (1.14)

$$\mathcal{T}' = \frac{\mathcal{T} \otimes \mathcal{A}^{N/L,-p/L}}{\mathbb{Z}_N} \tag{1.16}$$

and it is a fully consistent 3d TQFT. It captures the nontrivial correlation functions of the lines on the boundary. However, as we said above, $\mathcal{T}'$ is not "the theory on the boundary" except for $L = 1$. We will refer to it as "the effective boundary theory". We can think of the factor of $\mathcal{A}^{N/L,-p/L}$ as a 3d TQFT produced by the bulk so that the $\mathbb{Z}_N$ gauging can be performed.

We see that the 3d discussion of $\mathcal{T}'$ of (1.14) has a physical interpretation in terms of a 4d system with a boundary. We will discuss in detail the purely 3d system in Section 2 and the 4d interpretation in Section 3.

We will further generalize this discussion to interfaces between bulks with $p^+$ and $p^-$. Again, when $L^+ = L^- = 1$ there is a meaningful 3d theory on the interface. And for other values of $L^\pm$ there is only an effective description as above. It is

$$\frac{\mathcal{T} \otimes \mathcal{A}^{N/L^+,-p^+/L^+} \otimes \mathcal{A}^{N/L^-,p^-/L^-}}{\mathbb{Z}_N} \ . \tag{1.17}$$

As in the case of a boundary, the two factors of $\mathcal{A}^{N/L^\pm,\mp p^\pm/L^\pm}$ can be interpreted as being produced by the bulk in the two sides such that the $\mathbb{Z}_N$ gauging can be performed on the interface.

In Section 4, we review the bulk dynamics of the $SU(N)$ and the $PSU(N)$ gauge theories and discuss their interfaces. Here we use the results in Section 3 to construct the interfaces in the $PSU(N)$ theory by gauging the one-form $\mathbb{Z}_N$ symmetry of the corresponding interfaces in $SU(N)$ theory.

In several appendices we summarize some background information and extend the analysis in the body of the paper. Appendix A reviews the equivalence of different definitions of Abelian anyons and derives some useful facts we use in the paper. Appendix B reviews the properties of the Jacobi symbols that appear in the central charge of the minimal Abelian TQFT $\mathcal{A}^{N,p}$. In Appendix C, we demonstrate that every Abelian TQFT corresponds to a unitary chiral RCFT. In Appendix D, we prove the equivalence of different procedures that remove lines from a TQFT. Appendix E reviews and extends the analysis of a $\mathbb{Z}_N$ two-form gauge theory in 4d. In Appendix F, we generalize the discussion to a TQFT with an arbitrary Abelian one-form global symmetry group $\prod \mathbb{Z}_{N_I}$.

## 2 One-form symmetries in 3d and their gauging

### 2.1 One-form global symmetries in 3d TQFTs

In a 3d TQFT with a $\mathbb{Z}_N$ one-form symmetry, every line $W$ is in some $\mathbb{Z}_N$ representation of charge $q(W)$. This means that the line transforms under a symmetry group element $s$ by

$$a^s(\gamma)W(\gamma') = W(\gamma')e^{\frac{2\pi i s q(W)}{N}} \ , \tag{2.1}$$

---

[10]This quotient is related to the discussion in [14–18].

where the symmetry transformation is implemented by the symmetry line $a^s$ that braids with $W$ as illustrated in Figure 2 with $a$ the generating line of the symmetry. The charge $q(W)$ can be determined by the spins of the lines $h[W]$ [20] (for a later presentations see *e.g.* the mathematical treatment in [21] and a more physical review in [22])

$$q(W) = N\big(h[a] + h[W] - h[aW]\big) \bmod N \,, \tag{2.2}$$

where $aW$ denotes the unique line in the fusion of $a$ and $W$. (The line $aW$ is unique since $a$ is an Abelian anyon as explained in Appendix A.)

For the special case $W = a^{s'}$, the transformation under the group element $s$ is characterized by some integer $P \bmod N$

$$a^s(\gamma)a^{s'}(\gamma') = a^{s'}(\gamma')e^{-\frac{2\pi i s s' P}{N}} \,, \tag{2.3}$$

Using (2.2) we obtain

$$h[a^{s+s'}] - h[a^s] - h[a^{s'}] = \frac{Pss'}{N} \bmod 1 \,. \tag{2.4}$$

Consider the case $s' = -s$. Since particles and their antiparticles have the same spin $h[a^s] = h[a^{-s}] \bmod 1$, and $h[1] = 0 \bmod 1$, we find two solutions with a given $P \bmod N$

$$h[a^s] = \frac{ps^2}{2N} \bmod 1, \quad p \in \{0, 1, ..., 2N - 1\} \,, \tag{2.5}$$

with $p = P$ or $(P + N) \bmod 2N$.

The condition $a^N = 1$ in (1.6) requires that $a^N$ has spin $\frac{pN}{2} = 0 \bmod 1$ and hence $pN$ must be even. Therefore, for even $N$, the distinct cases are labeled by $p = 0, 1, ..., 2N - 1$ and for odd $N$, they are labeled by $p = 0, 2, ..., 2N - 2$.

Some different values of the label $p$ can be identified using group automorphisms. For a $\mathbb{Z}_N$ one-form symmetry, this amounts to choosing a new generating line for the symmetry $\widehat{a} = a^r$ with $\gcd(N, r) = 1$. The charge of a line $W$ in the TQFT becomes $q(W)r \bmod N$. The new generating line $\widehat{a}$ has spin $\frac{\widehat{p}}{2N} \bmod 1$ with $\widehat{p} = pr^2 \bmod 2N$ so the label $p$ and $\widehat{p} = pr^2$ mod $2N$ should be identified.

In a spin TQFT there are new elements. These theories include a transparent spin-half line $\psi$. Using the language of one-form symmetries, we can say that $\psi$ generates a $\mathbb{Z}_2$ one-from symmetry that does not act faithfully on the lines.

Consider first the case of even $N$. Here we can replace the generating line $a$ with $\widehat{a} = a\psi$, which also satisfies (1.6) $\widehat{a}^N = 1$. The spin of $\widehat{a}$ is $\frac{p}{2N} + \frac{1}{2} = \frac{p+N}{2N}$. Therefore, we can identify $p \sim p + N$. Equivalently, we can say that our system has a $\mathbb{Z}_N \otimes \mathbb{Z}_2$ one-form symmetry, where the first factor is generated either by $a$ or by $\widehat{a}$ and the second by $\psi$.

For odd $N$ we could contemplate $a^N = \psi$ and therefore allow odd $pN$ (and hence $p$ is also odd). This means that $a$ generates a $\mathbb{Z}_{2N}$ symmetry. Since $N$ is odd, $\mathbb{Z}_{2N} \cong \mathbb{Z}_N \otimes \mathbb{Z}_2$. Here, the first factor is generated by $\widehat{a} = a\psi$; indeed, $\widehat{a}^N = 1$. The second factor is generated by $\psi$. The $\mathbb{Z}_N$ factor is characterized by the label $\widehat{p} = (p + N) \bmod 2N$, which is even (because $p$ and $N$ are both odd). Therefore, without loss of generality, we can say that even in spin theories we impose that $pN$ is even. (Alternatively, we can allow odd $pN$, but identify $p \sim p + N$.)

The labels of distinct one-form symmetries for both non-spin and spin theories are summarized in Table 1. Recall that in addition, choosing a different generator for the $\mathbb{Z}_N$ symmetry changes $p$.

**Examples.** An example of a class of 3d TQFTs that has a $\mathbb{Z}_N$ one-form symmetry of all possible parameter $p = 0, \cdots, 2N - 1 \bmod 2N$ is the $U(1)_{pN}$ Chern-Simons theory. The symmetry lines

Table 1: The allowed labels $p$ for $\mathbb{Z}_N$ one-form symmetry up to the redundancy in redefining the generators of the symmetries. A $\mathbb{Z}_N$ one-form symmetry of parameter $p$ is generated by a line $a$ of spin $h[a] = \frac{p}{2N}$ mod 1. For a non-spin TQFT, we need $pN \in 2\mathbb{Z}$, and $p \sim p + 2N$. For a spin TQFT, we can use $pN \in \mathbb{Z}$ and $p \sim p + N$. Alternatively, we can say that in the spin case we keep the condition $pN \in 2\mathbb{Z}$ and add the identification $p \sim p + N$ only for even $N$.

|  | even $N$ | odd $N$ |
|---|---|---|
| non-spin TQFT | $p = 0, 1, ..., 2N-1$ | $p = 0, 2, ..., 2N-2$ |
| spin TQFT | $p = 0, 1, ..., N-1$ | $p = 0, 1, ..., N-1$ or equivalently $p = 0, 2, ..., 2N-2$ |

of the $\mathbb{Z}_N$ one-form symmetry are generated by the Wilson line $a$ of $U(1)$ charge $p$, and the line $a^s$ for a general group element $s$ has spin

$$h[a^s] = \frac{(ps)^2}{2pN} = \frac{ps^2}{2N} \quad \text{mod 1},$$ (2.6)

in accordance with (2.5).

Another example is the simplest Abelian $\mathbb{Z}_N$ gauge theory in 3d, denoted by $(\mathscr{Z}_N)_0$. The theory has a $\mathbb{Z}_N \times \mathbb{Z}_N$ one-form symmetry, generated by the basic electric and magnetic lines $V_E$, $V_M$ of integer spins. $V_E$ generates a $\mathbb{Z}_N$ one-form symmetry with $p = 0$ and $V_M$ generates another $\mathbb{Z}_N$ one-form symmetry with $p = 0$. However, these two lines $V_E$, $V_M$ have a mutual braiding phase $e^{-2\pi i/N}$. This fact can be used to find a $\mathbb{Z}_N \subset \mathbb{Z}_N \times \mathbb{Z}_N$ of arbitrary even label $p$. Specifically, the line

$$b = V_E^{p/2} V_M,$$ (2.7)

generates a $\mathbb{Z}_N \subset \mathbb{Z}_N \times \mathbb{Z}_N$ one-form symmetry and since its spin is $\frac{p}{2N}$ mod 1, the one-form symmetry is characterized by $p$.

What about the remaining lines? The line

$$c = V_E^{p/2} V_M^{-1},$$ (2.8)

generates a $\mathbb{Z}_N$ one-form symmetry of even parameter $-p$ mod $2N$. However, the lines $b$ and $c$ satisfy

$$(bc)^{N/\gcd(N,p)} = 1,$$ (2.9)

and therefore only when $\gcd(N, p) = 1$ do the two lines generate the entire $\mathbb{Z}_N \times \mathbb{Z}_N$ one-form symmetry.

Let us study a third example. We consider $U(1)_N \otimes U(1)_{-N}$ (for $N$ odd this is a spin TQFT) with gauge fields $z$ and $y$ and an action

$$\int \left( \frac{N}{4\pi} z dz - \frac{N}{4\pi} y dy \right).$$ (2.10)

Writing it in terms of $x = z - y$, this action becomes

$$\int \left( \frac{N}{4\pi} x dx + \frac{N}{2\pi} x dy \right),$$ (2.11)

and as in [4], it describes the $\mathbb{Z}_N$ DW theory [9] that we denote as $(\mathscr{Z}_N)_N$. It has a $\mathbb{Z}_N \times \mathbb{Z}_N$ one-form symmetry, generated by $Z = \exp(i \oint z)$ of spin $\frac{1}{2N}$ mod 1, and $Y = \exp(i \oint y)$ of

spin $-\frac{1}{2N}$ mod 1. The two lines $Z$ and $Y$ have trivial mutual braiding. The basic electric and magnetic lines of the DW $\mathbb{Z}_N$ gauge theory can be written as $V_E = ZY^{-1} = \exp(i \oint x)$ and $V_M = Y$. As in the previous example of $(\mathcal{Z})_0$, the line

$$b = Z^{(p+1)/2}Y^{-(p-1)/2} = V_E^{(p+1)/2}V_M \tag{2.12}$$

generates a $\mathbb{Z}_N \subset \mathbb{Z}_N \times \mathbb{Z}_N$ one-form symmetry of odd parameter $p \sim p + 2N$.

Again, we could ask about the remaining lines. The line

$$c = Z^{(p-1)/2}Y^{-(p+1)/2} = V_E^{(p-1)/2}V_M^{-1} \tag{2.13}$$

generates a $\mathbb{Z}_N$ one-form symmetry of odd parameter $-p$ mod $N$. As in the previous example, these lines satisfy a relation: $(bc)^{N/\gcd(N,p)} = 1$ and therefore only when $\gcd(N,p) = 1$ do the two lines $b$ and $c$ generate the entire $\mathbb{Z}_N \times \mathbb{Z}_N$ one-form symmetry.

Let us summarize the last two examples. A subset of the lines of $(\mathcal{Z}_N)_0$ generates a $\mathbb{Z}_N$ one-form symmetry with even parameter $p$ and a subset of the lines of $(\mathcal{Z}_N)_N$ generates a $\mathbb{Z}_N$ one-form symmetry with odd parameter $p$. When $\gcd(N,p) = 1$ the remaining lines also generate a $\mathbb{Z}_N$ one-form symmetry with parameter $-p$.

We can combine these two examples more concisely using the theory $(\mathcal{Z}_N)_{-pN}$ with the action

$$\int \left( -\frac{pN}{4\pi}xdx + \frac{N}{2\pi}xdy \right). \tag{2.14}$$

Here the parameter $p$ can be identified with $p + 2$ using the redefinition $y \to y - x$ so these theories are either $(\mathcal{Z}_N)_0$ or $(\mathcal{Z}_N)_N$, and the lines $b$ and $c$ in $(\mathcal{Z}_N)_0$ and $(\mathcal{Z}_N)_N$ are mapped to the following lines in $(\mathcal{Z}_N)_{-pN}$

$$b = \exp(i \oint y), \quad c = \exp(ip \oint x - i \oint y). \tag{2.15}$$

## 2.2 The minimal Abelian TQFT $\mathcal{A}^{N,p}$

In this section, we will show that when $\gcd(N,p) = 1$ and $pN \in 2\mathbb{Z}$ the $N$ symmetry lines associated to a $\mathbb{Z}_N$ one-form symmetry form a consistent TQFT. We call this theory "the minimal Abelian TQFT" and denote it by $\mathcal{A}^{N,p}$. This theory was first studied in [20] and more recently in [23, 24]. Here we emphasize its one-form global symmetry and show how it appears as a sub-theory in TQFTs with a $\mathbb{Z}_N$ one-form global symmetry.[11]

Using the assumed underlying $\mathbb{Z}_N$ one-form symmetry, we can simplify the discussion in [20]. The symmetry determines the spins of the lines $h[a^s] = \frac{ps^2}{2N}$ mod 1, and their braiding leads to the following $S$ matrix

$$S_{ss'} = \frac{1}{\sqrt{N}} \exp\left( 2\pi i \left( h[s] + h[s'] - h[ss'] \right) \right) = \frac{1}{\sqrt{N}} \exp\left( -\frac{2\pi ip}{N}ss' \right), \quad s,s' \in \{1,...,N\} . \tag{2.16}$$

This matrix is unitary only when $L = \gcd(N,p) = 1$. (If $L = \gcd(N,p) \neq 1$, the line $a^{N/L}$ has trivial braiding with all the lines in the theory, so the $S$ matrix is not unitary.)

The chiral central charge $c_N^{(p)}$ modulo 8 of the Abelian TQFT $\mathcal{A}^{N,p}$ can be computed using the following formula (see e.g. [22, 25])[12]

$$e^{i \frac{2\pi}{8} c_N^{(p)}} = \frac{1}{\sqrt{N}} \sum_{s=1}^{N} e^{2\pi ih[a^s]} . \tag{2.17}$$

---

[11]A putative theory with $N$ Abelian lines $a^s$ with $h(a^s) = \frac{ps^2}{2N}$ is not a consistent (modular) TQFT when $\gcd(N,p) \neq 1$.

[12]The chiral central charge of a TQFT can be shifted by adding a $(E_8)_1$ theory, since it has $c = 8$ and no nontrivial lines.

Table 2: The chiral central charge $c_N^{(p)}$ mod 8 of the minimal Abelian theory $\mathcal{A}^{N,p}$ computed from (2.18). For each case $c_N^{(p)}$ mod 8 is one of the two possible values depending on $[N] = N$ mod 4 and $[p] = p$ mod 4. Here $\times$ means that the theories with such $p$ and $N$ do not exist according to the conditions that $pN$ is even and $\gcd(N,p) = 1$.

| $[N]$ \ $[p]$ | 0 | 1 | 2 | 3 |
|---|---|---|---|---|
| 0 | $\times$ | 1,5 | $\times$ | 3,7 |
| 1 | 0,4 | $\times$ | 0,4 | $\times$ |
| 2 | $\times$ | 1,5 | $\times$ | 3,7 |
| 3 | 6,2 | $\times$ | 6,2 | $\times$ |

The summation is a Gaussian sum with the following closed-form expression [20, 26]

$$
\exp\left(i\frac{2\pi}{8}c_N^{(p)}\right) = \begin{cases} \left(\dfrac{p/2}{N}\right)\epsilon(N) & N \text{ odd}, p \text{ even} \\ \left(\dfrac{N/2}{p}\right)\epsilon(p)^{-1}\exp(\pi i/4) & N \text{ even}, p \text{ odd} \end{cases}, \tag{2.18}
$$

where $\epsilon(s) = 1$ for $s = 1$ mod 4, $\epsilon(s) = i$ for $s = -1$ mod 4 and $\left(\frac{a}{b}\right)$ is the Jacobi symbol reviewed in Appendix B. The values of the chiral central charges are summarized in Table 2, and they are always integers.

Every Abelian TQFT can be represented by some Abelian Chern-Simons theory [27–31] (for a review see *e.g.* [32]). It is also true for $\mathcal{A}^{N,p}$. For example,[13] $\mathcal{A}^{N,1} \longleftrightarrow U(1)_N$ and $\mathcal{A}^{N,N-1} \longleftrightarrow SU(N)_1$. An alternative description of $\mathcal{A}^{N,N-1}$ is the $U(1)^{N-1}$ Chern-Simons theory with the coefficient matrix given by the Cartan matrix of $SU(N)$ (see *e.g.* [25]). The dualities also hold after taking orientation-reversal.

Similar to one-form symmetries, any two minimal Abelian TQFTs $\mathcal{A}^{N,p}$ and $\mathcal{A}^{N,pr^2}$ with $\gcd(N,r) = 1$ are related by group automorphsims.

Following the discussion of spin TQFTs in the previous subsection we can generalize the minimal theory to spin theories. Originally, we imposed $a^N = 1$ and then $pN$ has to be even and the minimal theory is nonspin. We can make it into a spin TQFT by tensoring the almost trivial theory[14] $\{1, \psi\}$. After doing that, for odd $N$ we can further redefine $a \to a\psi$, which makes $a^N = \psi$ and shifts $p \to p + N$ making $pN$ odd. This way we can define a spin TQFT

$$
\mathcal{A}^{N,p} \equiv \mathcal{A}^{N,p+N} \otimes \{1, \psi\} \qquad \text{for odd } pN \text{ and } \gcd(N,p) = 1. \tag{2.19}
$$

This is the minimal spin TQFT generated by a line of spin $\frac{p}{2N}$ mod 1.

As an application, the spin TQFT $U(1)_N$ for odd $N$ factorizes[15]

$$
U(1)_N \longleftrightarrow \mathcal{A}^{N,N+1} \otimes \{1, \psi\}, \tag{2.20}
$$

---

[13]Typically (and perhaps always) the TQFT can be described by a Chern-Simons (CS) gauge theory and a corresponding Rational Conformal Field Theory (RCFT). In fact, there are often several distinct CS theories corresponding to the same TQFT. Then the symbol $\longleftrightarrow$ means that they are dual. It is important to stress, however, that distinct RCFTs with the same TQFT are often inequivalent.

[14]The almost trivial TQFT $\{1, \psi\}$ can be represented by $SO(M)_1$ for some integer $M$. The dependence on $M$ is only in the framing anomaly or equivalently in the chiral central charges $c = \frac{M}{2}$. See *e.g.* Appendix C of [33], Appendix B of [34], and also [6].

[15]We use equal sign to relate two isomorphic TQFTs. However, we used $\longleftrightarrow$ to denote two dual presentations of the same TQFT. Typically one or both of these presentations is given by a Chern-Simons gauge theory. Then the classical Chern-Simons theories are not equal (hence we do not use an equal sign), but the quantum theories are the dual.

where the first factor is a nonspin minimal theory. Since $\mathcal{A}^{N,N+1} = \mathcal{A}^{N,-N+1} \longleftrightarrow SU(N)_{-1}$. This reproduces the level-rank duality $U(1)_N \longleftrightarrow SU(N)_{-1}$, which is valid only as spin TQFTs [6].[16]

## 2.3 Factorization of 3d TQFTs when $\gcd(N,p) = 1$

In this section we show that a TQFT $\mathcal{T}$ with a $\mathbb{Z}_N$ one-form symmetry of label $p$ such that $\gcd(N,p) = 1$ factorizes as

$$\mathcal{T} = \mathcal{A}^{N,p} \otimes \mathcal{T}' \qquad \text{when } \gcd(N,p) = 1. \tag{2.21}$$

This is quite surprising. It means that in this case all the information about the global symmetry and its action on $\mathcal{T}$ is included in a decoupled factor of the minimal theory $\mathcal{A}^{N,p}$ and $\mathcal{T}'$ is invariant under the symmetry.[17]

The theory $\mathcal{T}$ includes the $\mathbb{Z}_N$ symmetry lines $a^s$. When $\gcd(N,p) = 1$ these lines form the minimal theory $\mathcal{A}^{N,p}$. Next, consider any line $W \in \mathcal{T}$. Since $a$ is Abelian, the fusion of $W$ with $a$ includes a single line rather than a sum of lines. (See Appendix A.) Therefore, since $\gcd(N,p) = 1$, we can always find an integer $s$ such that the line $W' = Wa^s$ has vanishing charge $q(W') = 0 \bmod N$. Denote the set of neutral lines $W'$ by $\mathcal{T}'$. This shows that every line $W \in \mathcal{T}$ is a product of a line $W' \in \mathcal{T}'$ and a line in $\mathcal{A}^{N,p}$. It is clear that all the conditions of a consistent TQFT are satisfied separately for $\mathcal{T}'$ and $\mathcal{A}^{N,p}$ and hence we have the factorization (2.21).

The factorization (2.21) also follows from a theorem in modular tensor category (see [16] and Theorem 3.13 in [17]). In physics language, the theorem states that if a 3d TQFT $\mathcal{T}$ has a consistent sub-theory $\mathcal{A}$, then $\mathcal{T}$ factorizes into $\mathcal{A} \otimes \mathcal{T}'$ where $\mathcal{T}'$ is another consistent TQFT that consists of all the lines in $\mathcal{T}$ that have trivial braiding with the lines in $\mathcal{A}$.[18]

Next, we use the fact that $(\mathcal{A}^{N,p} \otimes \mathcal{A}^{N,-p})/\mathbb{Z}_N$ is a trivial theory, where the quotient means gauging the anomaly free diagonal $\mathbb{Z}_N$ one-form symmetry generated by the two generating lines of the minimal Abelian TQFTs. This leads to an alternative presentation of the TQFT $\mathcal{T}'$

$$\mathcal{T}' = \frac{\mathcal{T} \otimes \mathcal{A}^{N,-p}}{\mathbb{Z}_N}, \tag{2.22}$$

where the quotient means gauging the anomaly free diagonal $\mathbb{Z}_N$ one-form symmetry generated by the symmetry generating line $a$ in $\mathcal{T}$ and the generating line of $\mathcal{A}^{N,-p}$.

Let us demonstrate this factorization in some examples.

The minimal Abelian TQFTs can be found as sub-theories in various examples discussed in Section 2.1. We start by considering $U(1)_{pN}$ when $\gcd(N,p) = 1$. The theory has a $\mathbb{Z}_{pN} \cong \mathbb{Z}_N \otimes \mathbb{Z}_p$ one-form symmetry with a $\mathbb{Z}_N$ subgroup generated by $a$, the Wilson line of charge $p$, and a $\mathbb{Z}_p$ subgroup generated by $b$, the Wilson line of charge $N$. The line $a$ and the line $b$ each generates a minimal Abelian TQFT $\mathcal{A}^{N,p}$ and $\mathcal{A}^{p,N}$. The full theory factorizes into these minimal Abelian TQFTs[19]

$$U(1)_{pN} \longleftrightarrow \mathcal{A}^{pN,1} = \mathcal{A}^{N,p} \otimes \mathcal{A}^{p,N} \text{ when } \gcd(N,p) = 1. \tag{2.23}$$

---

[16]If $N = 8n$ for some integer $n$, the non-spin minimal Abelian TQFT satisfies $\mathcal{A}^{N,1} = \mathcal{A}^{N,N+1}$ by redefining the generating line $a \to a^{4n+1}$. Thus $U(1)_{8n} \longleftrightarrow SU(8n)_{-1}$ are dual as non-spin TQFTs in agreement with [35].

[17]If the theory $\mathcal{T}$ is a spin TQFT, then since the transparent spin-half line is invariant under any one-form symmetry, the theory $\mathcal{T}'$ also contains such a line and is a spin TQFT.

[18]We thank Zhenghan Wang for discussions about this point.

[19]For odd $pN$ the full theory $U(1)_{pN}$ as well as $\mathcal{A}^{N,p}$ and $\mathcal{A}^{p,N}$ are spin TQFTs. The spin Chern-Simons theory $U(1)_{pN}$ can also factorize as $U(1)_{pN} \longleftrightarrow \mathcal{A}^{N,p+N} \otimes \mathcal{A}^{p,p+N} \otimes \{1, \psi\}$ (compare with (2.20)), where the first two factors are non-spin minimal theories.

To show the factorization of an Abelian TQFT, it is sufficient to check the factorization in the fusion rules, the spins of the lines and the chiral central charge. The fusion rules of $U(1)_{pN}$ are the same as the group law of $\mathbb{Z}_{pN}$. When $\gcd(N,p) = 1$, the group factorizes into $\mathbb{Z}_N \times \mathbb{Z}_p$ and every line in the theory can be decomposed into $W = a^s b^r$ with some unique $(s,r) \in \mathbb{Z}_N \times \mathbb{Z}_p$. The spins of the lines also factorize

$$h[W] = \frac{(ps+Nr)^2}{2pN} = \left( \frac{p}{2N}s^2 + \frac{N}{2p}r^2 \right) \bmod 1 = (h[a^s]+h[b^r]) \bmod 1. \qquad (2.24)$$

The chiral central charge of $U(1)_{pN}$ is $c = 1$. It agrees with the sum of the chiral central charges of individual sub-theories up to a periodicity of 8

$$e^{i\frac{2\pi}{8}\left(c_N^{(p)}+c_p^{(N)}\right)} = \frac{1}{\sqrt{N}} \sum_{j=0}^{N-1} e^{\frac{\pi i p j^2}{N}} \frac{1}{\sqrt{p}} \sum_{k=0}^{p-1} e^{\frac{\pi i N k^2}{p}} = \frac{1}{\sqrt{pN}} \sum_{j,k} e^{\frac{2\pi i (pj+Nk)^2}{2pN}} = e^{i\frac{2\pi}{8}}. \qquad (2.25)$$

We conclude that $U(1)_{pN}$ factorizes into $\mathcal{A}^{N,p} \otimes \mathcal{A}^{p,N}$ when $\gcd(N,p) = 1$.

The minimal Abelian TQFT $\mathcal{A}^{N,p}$ is also a sub-theory in $(\mathcal{Z}_N)_{-pN}$ when $\gcd(N,p) = 1$. Similarly, the theory also factorizes

$$(\mathcal{Z}_N)_{-pN} \longleftrightarrow \mathcal{A}^{N,p} \otimes \mathcal{A}^{N,-p} \text{ when } \gcd(N,p) = 1, \qquad (2.26)$$

where $\mathcal{A}^{N,p}$ and $\mathcal{A}^{N,-p}$ are generated by the lines $b$ and $c$ in $(\mathcal{Z}_N)_{-pN}$ defined in (2.15).

As a consistency check, combining (2.21) and (2.22) and using the factorization property of $(\mathcal{Z}_N)_{-pN}$ in (2.26), we recover the following canonical duality [36]

$$\mathcal{T} \longleftrightarrow \frac{\mathcal{T} \otimes (\mathcal{Z}_N)_{-pN}}{\mathbb{Z}_N} \longleftrightarrow \begin{cases} \dfrac{\mathcal{T} \otimes (\mathcal{Z}_N)_0}{\mathbb{Z}_N} & \text{even } p \\[2mm] \dfrac{\mathcal{T} \otimes (\mathcal{Z}_N)_N}{\mathbb{Z}_N} & \text{odd } p \end{cases}, \qquad (2.27)$$

where the quotient means gauging the anomaly free diagonal one-form symmetry generated by the line $a$ in $\mathcal{T}$ and the line $c$ in the $\mathbb{Z}_N$ gauge theories defined in (2.8), (2.13) and (2.15). The duality holds even when $\gcd(N,p) \neq 1$. Under the duality, the symmetry generating line $a$ in $\mathcal{T}$ is mapped to the line $b$ in the dual theories defined in (2.7), (2.12) and (2.15). Then the $\mathbb{Z}_N$ one-form symmetry is entirely in the $(\mathcal{Z}_N)_{-pN}$ factor.

We remark that although the 3d TQFT factorizes, the corresponding 2d RCFTs may not factorize since the unitary modular tensor category does not fully specify the 2d chiral conformal field theory [37]. For Abelian TQFTs, we provide a construction of a corresponding unitary chiral RCFT in Appendix C.

## 2.4 't Hooft anomaly of one-form global symmetries

Consider a 3d TQFT $\mathcal{T}$ with a $\mathbb{Z}_N$ one-form symmetry of label $p$ with the symmetry generating line $a$. Gauging the one-form symmetry amounts to summing over all possible insertions of the symmetry lines [1]. If the symmetry lines have non-integer spin, the partition function vanishes because of the summation. This means that the one-form symmetry has an 't Hooft anomaly unless $p = 0 \bmod 2N$. Indeed, the one-form symmetry of label $p = 0 \bmod 2N$ can be gauged following the procedure outline in Section 1. (When $p = N$, the theory can also be gauged as a spin TQFT by redefining the symmetry generating line using the transparent spin-half line. After gauging it becomes a spin TQFT, even though the original theory can be a non-spin theory. It reflects a mixed 't Hooft anomaly between the one-form symmetry and gravity, which we will explain in details later.)

We couple the one-form symmetry of the 3d TQFT to a classical $\mathbb{Z}_N$ two-form gauge field $\mathcal{B}_{\mathcal{C}} \in H^2(\mathcal{M}_4, \mathbb{Z}_N)$.[20] The anomaly of the one-form symmetry is characterized by a 4d term of the gauge field $\mathcal{B}_{\mathcal{C}}$ through anomaly inflow. To determine the 4d term, we use the canonical duality in (2.27) [36]

$$\mathcal{T} \longleftrightarrow \frac{\mathcal{T} \otimes (\mathcal{Z}_N)_{-pN}}{\mathbb{Z}_N} \, . \tag{2.28}$$

Under the duality, the original $\mathbb{Z}_N$ one-form symmetry in $\mathcal{T}$ is mapped to the one-form symmetry generated by line $b$ defined in (2.15) in the dual description so the theory on the right hand side couples to the classical gauge field $\mathcal{B}_{\mathcal{C}}$ through the $(\mathcal{Z}_N)_{-pN}$ factor. It was shown in [4] that the anomaly of $(\mathcal{Z}_N)_{-pN}$ is cancelled by the 4d term

$$2\pi \frac{p}{2N} \int_{\mathcal{M}_4} \mathcal{P}(\mathcal{B}_{\mathcal{C}}) \, , \tag{2.29}$$

where $\mathcal{P}$ is the Pontryagin square operation (for a review see e.g. [10,38,39]). Therefore, the anomaly of a $\mathbb{Z}_N$ one-form symmetry of label $p$ is characterized by the 4d term (2.29) [1,4,39].

The 4d term (2.29) is consistent with the $\mathbb{Z}_N$ periodicity of the $\mathcal{B}_{\mathcal{C}}$ field only for even $pN$. Furthermore, for $p = N$ (which is possible only for even $N$) it can be written as

$$\pi \int_{\mathcal{M}_4} \mathcal{P}(\mathcal{B}_{\mathcal{C}}) = \left( \pi \int_{\mathcal{M}_4} \mathcal{B}_{\mathcal{C}} \cup \mathcal{B}_{\mathcal{C}} \right) \bmod 2\pi = \left( \pi \int_{\mathcal{M}_4} w_2(\mathcal{M}_4) \cup \mathcal{B}_{\mathcal{C}} \right) \bmod 2\pi \, , \tag{2.30}$$

where $w_2(\mathcal{M}_4) \in H^2(\mathcal{M}_4, \mathbb{Z}_2)$ is the second Stiefel-Whitney classe of the manifold $\mathcal{M}_4$ (see e.g. [40, 41]). Equation (2.30) follows from the identity $x \cup x = w_2(\mathcal{M}_4) \cup x$ for $x = (\mathcal{B}_{\mathcal{C}} \bmod 2) \in H^2(\mathcal{M}_4, \mathbb{Z}_2)$ (on orientable manifolds). We interpret the 4d term (2.30) as a mixed 't Hooft anomaly between the one-form symmetry and gravity (fermion parity), which means that when this anomaly exists the one-form symmetry can be gauged only on spin manifolds. See also the related discussion in appendix E.

On spin manifolds, $pN$ in (2.29) can be odd. Furthermore, (2.29) vanishes for $p = N$.

In summary, on non-spin manifolds, the anomaly is labeled by $p = 0, 1, ..., 2N-1$ for even $N$ and $p = 0, 2, ..., 2N-2$ for odd $N$, and on a spin manifolds, the anomaly is labeled by $p = 0, 1, ..., N-1$. This agrees with the labels of 3d $\mathbb{Z}_N$ one-form symmetries listed in Table 1.

The anomaly can be changed by choosing a different generating line $\widehat{a} = a^r$ with $\gcd(N, r) = 1$ as explained in Section 2.1. It is equivalent to redefining the classical gauge field $\mathcal{B}_{\mathcal{C}}$ by a multiplication by $r$ and the anomaly coefficient in (2.29) becomes $pr^2 \bmod 2N$.

In the presence of the classical gauge field $\mathcal{B}_{\mathcal{C}}$, the line $W$ is dressed with an open surface $e^{-\frac{2\pi i q(W)}{N} \int \mathcal{B}_{\mathcal{C}}}$ for gauge invariance and the redefinition of the classical gauge field $\mathcal{B}_{\mathcal{C}}$ rescales the charge from $q(W)$ to $q(W)r$.

An anomalous $\mathbb{Z}_N$ one-form symmetry can have anomaly free subgroups. On spin manifolds, a $\mathbb{Z}_m$ subgroup is anomaly free if the symmetry generator $\widehat{a} = a^{N/m}$ has integer or half-integer spin

$$h[\widehat{a}] = \frac{pN}{2m^2} \in \frac{1}{2}\mathbb{Z} \, . \tag{2.31}$$

There is always a $\mathbb{Z}_L$ subgroup with $L = \gcd(N, p)$ that satisfies this condition and hence it is anomaly free. But the $\mathbb{Z}_L$ subgroup may not be the maximal anomaly free subgroup. For $NL = r^2 t$ with some integers $r, t$ such that $t$ does not contain any complete-square divisors great than one, the maximal anomaly free subgroup is $\mathbb{Z}_r$. As a non-spin TQFT, a $\mathbb{Z}_m$ subgroup is anomaly free only if $h[\widehat{a}] \in \mathbb{Z}$ and therefore the $\mathbb{Z}_L$ subgroup is anomaly free only for even $pN/L^2$.

---

[20]The subscript $\mathcal{C}$, as in $B_{\mathcal{C}}$, denotes that the gauge field is classical.

## 2.5 A generalization of the three-step gauging procedure to anomalous theories

In this subsection, we will introduce a new operation on 3d TQFTs that generalizes the three-step gauging procedure outlined in Section 1. This generalized operation will appear naturally in Section 3, where we consider 4d theories with boundaries and interfaces.

The standard gauging procedure of an anomaly free $\mathbb{Z}_N$ one-form symmetry can be used when $p = 0$, where all the symmetry lines have integer spins. Then in step 1 we remove the non-invariant lines, in step 2 we identify lines that differ by the fusion with the symmetry lines, and in step 3 we take lines at fixed points of the identification several times.

When $p = N$ this simple process cannot be repeated because the generating line $a$ has half-integer spin. As we said above, this can be interpreted as a mixed anomaly between the one-form symmetry and gravity. This anomaly vanishes on spin manifolds and therefore, we can gauge the symmetry and find a spin TQFT. Let us discuss it in more detail. If the original TQFT $\mathcal{T}$ is a spin theory, it has a transparent spin-half line $\psi$. Otherwise, we make it into a spin TQFT by tensoring the almost trivial theory $\{1, \psi\}$. Now that we have a spin TQFT we can redefine $a \rightarrow \widehat{a} = a\psi$. Since $p = N$ and $pN$ is even, this occurs only for even $N$ and then the redefinition preserves the fact that $a^N = 1$. The redefinition shifts $p$ to be zero. As a result, even in this case we can use the standard three-step gauging process with $\widehat{a}$. The only difference is that the theory is spin.

For simplicity from this point on we will limit ourselves to spin TQFTs.

Consider a 3d spin TQFT $\mathcal{T}$ with a $\mathbb{Z}_N$ one-form symmetry of label $p$ such that $\gcd(N, p) = 1$, the spin TQFT factorizes as discussed in Section 2.3

$$\mathcal{T} = \mathcal{T}' \otimes \mathcal{A}^{N,p}, \tag{2.32}$$

where $\mathcal{T}'$ is the 3d spin TQFT that consists of all the $\mathbb{Z}_N$ invariant lines in $\mathcal{T}$, and it can be extracted through

$$\mathcal{T}' = \frac{\mathcal{T} \otimes \mathcal{A}^{N,-p}}{\mathbb{Z}_N}. \tag{2.33}$$

In this case, we define an operation that maps $\mathcal{T}$ to $\mathcal{T}'$. The operation discards all the $\mathbb{Z}_N$ non-invariant lines in $\mathcal{T}$. It is equivalent to applying only the step 1 of the three-step gauging procedure.

When $\gcd(N, p) \neq 1$, the $\mathbb{Z}_N$ one-form symmetry has an anomaly free $\mathbb{Z}_L$ subgroup generated by $\widehat{a} = a^{N/L}$ with $L = \gcd(N, p)$. Gauging this $\mathbb{Z}_L$ subgroup produces a new spin TQFT $\mathcal{T}/\mathbb{Z}_L$. The new spin TQFT contains the original symmetry generating line $a$, but now it generates a $\mathbb{Z}_{N'}$ one-form symmetry ($N' = N/L$) with label $p' = p/L$. Since $\gcd(N', p') = 1$, $\mathcal{T}/\mathbb{Z}_L$ factorizes

$$\frac{\mathcal{T}}{\mathbb{Z}_L} = \left(\frac{\mathcal{T}}{\mathbb{Z}_L}\right)' \otimes \mathcal{A}^{N/L, p/L}, \tag{2.34}$$

where $(\mathcal{T}/\mathbb{Z}_L)'$ contains all the lines in $\mathcal{T}/\mathbb{Z}_L$ that have trivial braiding with $a$. We define the generalized gauging operation that maps

$$\mathcal{T} \rightarrow \mathcal{T}' \equiv \left(\frac{\mathcal{T}}{\mathbb{Z}_L}\right)' = \frac{\mathcal{T}/\mathbb{Z}_L \otimes \mathcal{A}^{N/L, -p/L}}{\mathbb{Z}_{N/L}} = \frac{\mathcal{T} \otimes \mathcal{A}^{N/L, -p/L}}{\mathbb{Z}_N}. \tag{2.35}$$

In both presentations, the quotient in the denominator uses the symmetry generator $a$ and the generating line of the minimal Abelian TQFT. In the second presentation, the $\mathbb{Z}_L$ subgroup of the $\mathbb{Z}_N$ quotient acts only on $\mathcal{T}$.

There are three ways to think about the map (2.35).

First, as we motivated it and as in the first presentation in (2.35), we first gauge the $\mathbb{Z}_L$ subgroup of the $\mathbb{Z}_N$ one-form symmetry and then remove the sub-theory in $\mathcal{T}/\mathbb{Z}_L$ consisting of the $\mathbb{Z}_{N/L}$ symmetry lines.

Second, since the $\mathbb{Z}_N$ symmetry is anomalous, we tensor a minimal theory $\mathcal{A}^{N/L,-p/L}$ that cancels the anomaly and then gauge the new anomaly free $\mathbb{Z}_N$ symmetry. This is clear in the second presentation in (2.35).

Third, we can perform step 1 of the three-step gauging procedure using the full $\mathbb{Z}_N$ symmetry and then perform steps 2 and 3 using only its $\mathbb{Z}_L$ subgroup:

Step 1  Select the lines invariant under the $\mathbb{Z}_N$ one-form symmetry. In particular, among the symmetry lines, only the ones associated to the $\mathbb{Z}_L$ subgroup generated by $\widehat{a} = a^{N/L}$ remain.

Step 2  If $\widehat{a}$ has integer spin, identify $W \sim W\widehat{a}$ and if $\widehat{a}$ has half-integer spin, identify $W \sim W\widehat{a}\psi$.

Step 3  Take multiple copies at the fixed points of the identification.

When $p = 0, N$, the symmetry is anomaly free and the generalized gauging operation reduces to the standard gauging procedure that produces $\mathcal{T}' = \mathcal{T}/\mathbb{Z}_N$.

In general, the $\mathbb{Z}_N$ one-form symmetry can have larger anomaly free $\mathbb{Z}_m$ subgroups that contain the $\mathbb{Z}_L$ subgroup. In Appendix D we show that the same result (2.35) can be reproduced if we first gauge the $\mathbb{Z}_m$ subgroup and then apply the generalized gauging operation to the remaining theory (up to a possible transparent spin-half line, which we will ignore).

Below we will see similar operations on TQFTs, which are not minimal. Following the second presentation in (2.35), we can tensor not the minimal theory $\mathcal{A}^{N/L,-p/L}$, but other theories that cancel the anomaly, e.g.

$$\frac{\mathcal{T} \otimes \mathcal{A}^{N/L^+,-p^+/L^+} \otimes \mathcal{A}^{N/L^-,p^-/L^-}}{\mathbb{Z}_N}, \tag{2.36}$$

where $p = p^+ - p^-$ and $L^{\pm} = \gcd(N, p^{\pm})$. The operation adds to the theory $\mathcal{T}$ two minimal Abelian TQFTs to cancel the anomaly and then gauges the diagonal one-form symmetry. The two minimal Abelian TQFTs $\mathcal{A}^{N/L^+,-p^+/L^+} \otimes \mathcal{A}^{N/L^-,-p^-/L^-}$ always have greater or equal number of lines than $\mathcal{A}^{N/L,-p/L}$ with $L = \gcd(N, p)$ and $p = p^+ - p^-$.[21] All the lines in $\mathcal{T}'$ defined in (2.35) can be identified with the lines from the original TQFT $\mathcal{T}$. In contrast, the theory (2.36) in general has additional lines.

# 3  Coupling to a 4d bulk

## 3.1  The bulk coupling

Consider a 4d symmetry protected topological (SPT) phase of $\mathbb{Z}_N$ one-form symmetry with the same action as the anomaly in (2.29)

$$2\pi \frac{p}{2N} \int_{\mathcal{M}_4} \mathcal{P}(\mathcal{B}_{\mathcal{C}}), \tag{3.1}$$

where $\mathcal{B}_{\mathcal{C}} \in H^2(\mathcal{M}_4, \mathbb{Z}_N)$ is a classical $\mathbb{Z}_N$ two-form gauge field. The theory has a description, reviewed in Appendix E, in terms of a dynamical $U(1)$ one-form gauge field $A$ and a classical $U(1)$ two-form gauge field $B_{\mathcal{C}}$

$$\int_{\mathcal{M}_4} \left( \frac{pN}{4\pi} B_{\mathcal{C}} B_{\mathcal{C}} + \frac{N}{2\pi} B_{\mathcal{C}} dA \right). \tag{3.2}$$

---

[21]$\mathcal{A}^{N/L,-p/L}$ has $N/L$ lines and $\mathcal{A}^{N/L^+,-p^+/L^+} \otimes \mathcal{A}^{N/L^-,p^-/L^-}$ has $N^2/L^+L^-$ lines. The ratio between them is $\frac{NL}{L^+L^-} = \left(\frac{N\ell}{L^+L^-}\right)\left(\frac{L}{\ell}\right)$ with $\gcd(N, p^+, p^-) = \ell$. Since the two factors are integers, the product theory has more lines.

The equation of motion of $A$ constrains $B_\mathcal{C}$ to be a $\mathbb{Z}_N$ two-form gauge field $\frac{2\pi}{N}\mathcal{B}_\mathcal{C}$.

The theory (3.2) is invariant under a one-form gauge transformation of background fields

$$B_\mathcal{C} \to B_\mathcal{C} - d\lambda, \quad A \to A + p\lambda, \tag{3.3}$$

with $\lambda$ a one-form gauge parameter.

We put the theory on a 4-manifold $\mathcal{M}_4$ with a boundary.[22] The action is gauge invariant under (3.3) up to a boundary term

$$-\int_{\partial\mathcal{M}_4} \left( \frac{pN}{4\pi}\lambda d\lambda + \frac{N}{2\pi}\lambda dA \right). \tag{3.4}$$

It can be cancelled by a theory on the boundary with a $\mathbb{Z}_N$ one-form symmetry of anomaly $p$, that couples to the classical gauge field $B_\mathcal{C}$. So we are going to place on the boundary an arbitrary TQFT $\mathcal{T}$ with such a symmetry and anomaly.

The coupling of the boundary TQFT $\mathcal{T}$ to the classical gauge field $B_\mathcal{C}$ has a convenient Lagrangian description using the canonical duality in (2.27) [36]

$$\mathcal{T} \quad \longleftrightarrow \quad \frac{\mathcal{T} \otimes (\mathscr{Z}_N)_{-pN}}{\mathbb{Z}_N} \tag{3.5}$$

and the Lagrangian description (2.14) of the second factor in the numerator. Then the classical gauge field $B_\mathcal{C}$ couples to the boundary theory through the $(\mathscr{Z}_N)_{-pN}$ theory

$$\int_{\partial\mathcal{M}_4} \left( -\frac{pN}{4\pi}x dx + \frac{N}{2\pi}x dy + \frac{N}{2\pi}B_\mathcal{C} y - \frac{N}{2\pi}B_\mathcal{C} A \right), \tag{3.6}$$

where the last term $B_\mathcal{C}A$ can be absorbed into the bulk action by modifying $B_\mathcal{C}dA$ to $AdB_\mathcal{C}$. Now the one-form gauge transformation (3.3) acts as

$$B_\mathcal{C} \to B_\mathcal{C} - d\lambda, \quad A \to A + p\lambda, \quad x \to x + \lambda, \quad y \to y + p\lambda. \tag{3.7}$$

## 3.2 Gauge the one-form symmetry

The whole system is anomaly free so there is no obstruction to gauging the one-form symmetry by turning the background gauge field $B_\mathcal{C}$ into a dynamical gauge field denoted by $B$. After gauging, the bulk theory becomes a dynamical $\mathbb{Z}_N$ two-form gauge theory reviewed in Appendix E. For later convenience, we define

$$L = \gcd(N, p), \quad K = N/L. \tag{3.8}$$

The bulk $\mathbb{Z}_N$ two-form gauge theory is effectively a $\mathbb{Z}_L$ one-form gauge theory[23] [1,4]. It has $L$ genuine line operators generated by $V = \exp(iK \oint A)$ and $L$ surface operators generated by $U = \exp(i \oint B)$. We will be interested in the effect of gauging on the boundary TQFT. For simplicity, we will limit ourselves to spin 4-manifolds.

It is important to stress that when $L \neq 1$ the bulk theory is nontrivial and hence it is meaningless to ask what the 3d theory on the boundary is. Instead, it should be thought of as part of the 4d-3d system. Nevertheless, we can discuss the physical observables such as the

---

[22]We restrict to the 4-manifolds such that every $\mathbb{Z}_N$ two-form gauge field on the boundary can be extended to the bulk. It requires the third relative cohomology $H^3(\mathcal{M}_4, \partial\mathcal{M}_4; \mathbb{Z}_N)$ to vanish.

[23]When $pN/L^2$ is odd, $V = \exp(iK \oint A)$ represents the worldline of a fermionic particle and the bulk theory is effectively a $\mathbb{Z}_L$ gauge theory that couples to $w_2(\mathcal{M}_4)$ of the manifold (see Appendix E).

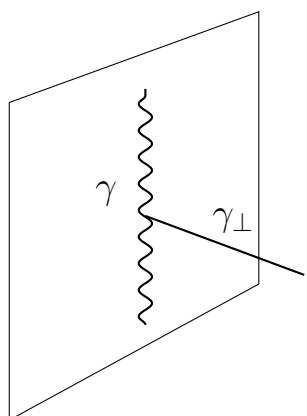

Figure 3: If a boundary line $W(\gamma)$ is at the fixed point of the identification using $\hat{a} = a^K$, it can form a junction by emanating a bulk line $\exp(iK \int_{\gamma_\perp} A)$.

line operators on the boundary and their correlation functions. We will extract from the 4d-3d system an effective boundary theory that reproduces many of these observables.

Let us examine the line operators on the boundary. The bulk $\mathbb{Z}_N$ gauge theory has $L$ line operators. When they are restricted to the boundary, they are regarded as boundary lines. But they have trivial braiding with all the boundary lines since they can smoothly move into the bulk and get un-braided. This means that unless $L = 1$ (where the bulk is trivial) the boundary lines to do not form a modular TQFT [4].

What are the other lines on the boundary? They can be constructed by fusing a line $W$ from the 3d TQFT $\mathcal{T}$ and the bulk lines generated by $\exp(i \oint A)|$, where | denotes the restriction to the boundary

$$W(\gamma) \exp\left(im \oint_\gamma A\right) \exp\left(i(mp - q(W)) \int_\Sigma B\right) \qquad \text{with} \qquad \gamma = \partial \Sigma \,, \qquad (3.9)$$

where $m \sim m + N$. The coupling to $B$ is needed for the one-form gauge symmetry. Next, we impose that these lines are genuine line operators, i.e. independent of the choice of surface $\Sigma$. This happens when $q(W) = mp$ mod $N$ [1, 4]

$$W(\gamma) \exp\left(im \oint_\gamma A\right) \qquad \text{with} \qquad q(W) = mp \text{ mod } N \,. \qquad (3.10)$$

An operator $W$ for which we cannot solve $q(W) = mp$ mod $N$ cannot be "dressed" to a physical line operator. In addition, using (3.6), the equation of motion of $B$ on the boundary leads to

$$\exp(i \oint A)| = \exp(i \oint y) \,. \qquad (3.11)$$

Now, the canonical duality (2.27) maps $\exp(i \oint y)$ to the symmetry generating line $a \in \mathcal{T}$, so $\exp(i \oint A)| = a$. Therefore all the line operators on the boundary are the $\mathbb{Z}_N$-invariant lines in $\mathcal{T}$. This means that we have performed only step 1 of the three-step gauge procedure.

Using this identification we also recognize the $L$ symmetry lines associated to the $\mathbb{Z}_L$ subgroup generated by $\hat{a} = a^K$ as the bulk lines generated by $V = \exp(iK \oint A)$. As we said above, these lines have trivial braiding with all the lines on the boundary.

One of the main points in our discussion is that since the bulk lines are trivial in any 3d correlation functions, we find it natural to identify them with the trivial line and accordingly,

Table 3: Gauging an SPT phase of $\mathbb{Z}_N$ one-form symmetry with a boundary supporting a 3d TQFT $\mathcal{T}$ leads to a 4d-3d system. It is not meaningful to discuss the resulting boundary theory unless the bulk is trivial. This happens when $L = \gcd(N, p) = 1$. However, we can extract an effective boundary theory that captures many of the features for any $L$.

| gauging | bulk | boundary theory | effective boundary theory |
|---|---|---|---|
| none | SPT of $\mathbb{Z}_N$ | $\mathcal{T}$ | |
| $\mathbb{Z}_N$ with $L=1$ | trivial | $\mathcal{T}' = \dfrac{\mathcal{T} \otimes \mathcal{A}^{N,-p}}{\mathbb{Z}_N}$ | |
| $\mathbb{Z}_N$ with $L \neq 1$ | $\mathbb{Z}_L$ gauge theory | not meaningful | $\mathcal{T}' = \dfrac{\mathcal{T} \otimes \mathcal{A}^{N/L,-p/L}}{\mathbb{Z}_N}$ |

identify the boundary lines $W \sim W\widehat{a}$. This works when $pN/L^2$ is even, so that the bulk line $V| = \widehat{a} = a^K$ has integer spin $h[\widehat{a}] = pN/2L^2$. When $pN/L^2$ is odd, the bulk line $V$ is charged under the $\mathbb{Z}_2$ fermion parity (see Appendix E), and on the boundary it is identified with $\widehat{a}$ of half-integer spin. Thus, we identify $W \sim W\widehat{a}\psi$. The procedure above is equivalent to quotienting by the boundary lines that can move to the bulk. This is essentially the step 2 of the gauging procedure, except that we perform it with respect to $\mathbb{Z}_L$ rather than with respect to $\mathbb{Z}_N$. As with the step 3 in the gauging procedure, the identification leads to new lines. Consider a boundary line $W$ at the fixed point of the fusion with $\widehat{a}$. It can form junctions by emanating bulk lines at some points as shown in Figure 3. When the bulk lines are viewed as trivial, these junctions become new boundary line operators.

We have just performed step 1 of the gauging with respect to $\mathbb{Z}_N$ and steps 2 and 3 with respect to its $\mathbb{Z}_L$ subgroup. The result is exactly $\mathcal{T}'$ defined in (2.35)

$$\mathcal{T}' = \frac{\mathcal{T} \otimes \mathcal{A}^{N/L,-p/L}}{\mathbb{Z}_N} \, . \tag{3.12}$$

We note that the identification by the bulk lines, whose correlation functions on the boundary are trivial, is similar to the procedure of the more mathematical analysis in [14–18].

In this system, the minimal theory $\mathcal{A}^{N/L,-p/L}$ can be interpreted as the 3d TQFT that the bulk theory provides to cancel the anomaly.

After gauging the $\mathbb{Z}_N$ one-form symmetry, there is an emergent dual $\mathbb{Z}_N$ one-form symmetry in the bulk and an emergent dual $\mathbb{Z}_N$ zero-form symmetry on the boundary. They are both generated by $\exp(i \oint B)$. The original system can be recovered by gauging these emergent symmetries.

In summary, starting with a general 3d TQFT $\mathcal{T}$ with a $\mathbb{Z}_N$ one-form symmetry of anomaly $p$, by coupling it to the bulk (3.1) and then gauging the one-form symmetry, we find the 3d TQFT $\mathcal{T}'$ as the effective boundary theory. We emphasize again that $\mathcal{T}'$ is only an effective theory, since the boundary can only be thought of as part of the 4d-3d system when the bulk theory is nontrivial. However, in the special cases when $L = 1$, the bulk theory is trivial and $\mathcal{T}'$ is the theory on the boundary.

### 3.3 Interfaces between two different bulk TQFTs

A generalization[24] is to consider interfaces between two different SPT phases of $\mathbb{Z}_N$ one-form symmetry one with coefficient $p^+$ and the other with $p^-$

$$S_{4d} = \int_{\mathcal{M}_4^-} \left( \frac{p^- N}{4\pi} B_{\mathcal{C}}^- B_{\mathcal{C}}^- + \frac{N}{2\pi} B_{\mathcal{C}}^- dA^- \right) + \int_{\mathcal{M}_4^+} \left( \frac{p^+ N}{4\pi} B_{\mathcal{C}}^+ B_{\mathcal{C}}^+ + \frac{N}{2\pi} B_{\mathcal{C}}^+ dA^+ \right) . \tag{3.13}$$

On the interface $\partial \mathcal{M}_4^+ = \partial \mathcal{M}_4^-$, we choose the boundary condition $B_{\mathcal{C}} = B_{\mathcal{C}}^+| = B_{\mathcal{C}}^-|$ where $|$ represents the restriction to the interface. The anomaly inflow can be cancelled by an interface theory with a $\mathbb{Z}_N$ one-form symmetry of anomaly $p = p_+ - p_-$ that couples to $B_{\mathcal{C}}$. As in the case of a boundary, which we discussed above, we place on the interface a 3d TQFT $\mathcal{T}$ with a $\mathbb{Z}_N$ one-form symmetry generated by $a$ and with anomaly $p$. Following the discussion of the boundary, we use the canonical duality (2.27) and couple the interface theory to $B_{\mathcal{C}}$ through the $(\mathcal{Z}_N)_{-pN}$ factor

$$S_{3d} = \int_{\partial \mathcal{M}_4} \left( -\frac{pN}{4\pi} x dx + \frac{N}{2\pi} x dy + \frac{N}{2\pi} B_{\mathcal{C}} y - \frac{N}{2\pi} B_{\mathcal{C}} (A^+ - A^-) \right) . \tag{3.14}$$

The one-form gauge symmetry of the system is

$$B_{\mathcal{C}}^\pm \to B_{\mathcal{C}}^\pm - d\lambda, \quad A^\pm \to A^\pm + p^\pm \lambda, \quad x \to x + \lambda, \quad y \to y + p\lambda . \tag{3.15}$$

We can gauge the $\mathbb{Z}_N$ one-form symmetry in the full system, i.e. make $B_{\mathcal{C}}$ dynamical (and remove the subscript $\mathcal{C}$). For later convenience, we define

$$L^\pm = \gcd(N, p^\pm), \quad L = \gcd(L^+, L^-), \quad K^\pm = N/L^\pm, \quad K = N/L = \mathrm{lcm}(K^+, K^-) . \tag{3.16}$$

After gauging, the bulk theory becomes effectively a $\mathbb{Z}_{L^\pm}$ one-form gauge theory on each side. In the special cases when $L^\pm = 1$, the bulk theories on both sides are trivial and there is a meaningful 3d theory on the interface. Otherwise, the interface can only be thought of as coupled to the 4d TQFT.

All the line operators $\widetilde{W}$ on the interface can be constructed by fusing the lines $W$ from the original 3d TQFT $\mathcal{T}$ and the lines $\widehat{V}_\pm = \exp(i \oint A^\pm)|$

$$\widetilde{W} = W \widehat{V}_+^{m^+} \widehat{V}_-^{m^-} , \qquad q(W) = (p^+ m^+ + p^- m^-) \bmod N . \tag{3.17}$$

The various factors in $\widetilde{W}$ are not $\mathbb{Z}_N$ gauge invariant line operators – each of them needs to be attached to a surface with $B$ to make them invariant. But the condition on $m^\pm$ means that their product $\widetilde{W}$ is $\mathbb{Z}_N$ invariant and hence it is a genuine line operator. (We ignore here a possible trivial open surface $\exp(iN \int B)$ and use $m^\pm \sim m^\pm + N$.) An operator $W$ for which we cannot solve this equation cannot be "dressed" to a physical operator.

Using the equation of motion of $B$

$$\widehat{V}_+ = a \widehat{V}_- \tag{3.18}$$

and that $a$ is a special case of $W$, all the lines on the interface can be written as[25]

$$\widetilde{W} = W \widehat{V}_-^{m^-} , \qquad q(W) = p^- m^- \bmod N . \tag{3.19}$$

---

[24]As above, for simplicity, we will limit ourselves to spin 4-manifolds.

[25]Note that unlike the case of a boundary discussed in Section 3.2, where $a = \widehat{V}$ was a line in the original theory $\mathcal{T}$, here the interface lines with $m^- \neq 0$ were not present in $\mathcal{T}$. Correspondingly, there are new interface lines that arise from the bulk degrees of freedom.

Table 4: The emergent global symmetry in a 4d system with an interface. The first row summarizes the symmetries and their spontaneous breaking. The second row presents the charge generators. In $U$ the integral is over a closed surface that pierces the interface. $U_\pm^{L^\pm} = 1$ means that this symmetry is broken to $\mathbb{Z}_{K^\pm}$. Below we will study an effective theory on the interface by performing a quotient of the full 4d-3d system by the bulk modes. We will see that the one-form global symmetry of this effective theory is $\mathbb{Z}_{\gcd(K^+,K^-)} = \mathbb{Z}_{N/\mathrm{lcm}(L^+,L^-)}$.

| bulk at $\mathcal{M}_4^-$ | Interface | bulk at $\mathcal{M}_4^+$ |
|---|---|---|
| one-form: $\mathbb{Z}_N \to \mathbb{Z}_{K^-}$ | one-form: $\mathbb{Z}_N \to 1$ | one-form: $\mathbb{Z}_N \to \mathbb{Z}_{K^+}$ |
| $U_- = \exp(i \oint B^-)$ | $U = \exp(i \int B^+ - i \int B^-)$ | $U_+ = \exp(i \oint B^+)$ |

Let us discuss the global symmetry of the system and its breaking (Table 4). After gauging, the bulk theories have an emergent $\mathbb{Z}_N$ one-form symmetry. It is spontaneously broken to $\mathbb{Z}_{K^\pm}$ on the two sides. The broken $\mathbb{Z}_{L^\pm} = \mathbb{Z}_N/\mathbb{Z}_{K^\pm}$ one-form symmetry is generated by the surface operator $U_\pm = \exp(i \oint B^\pm)$ with $U_\pm^{L^\pm} = 1$. It acts on the $\mathbb{Z}_{L^\pm}$ gauge theories in the two sides.

The interface has an emergent symmetry generated by the surface operator that pierces the interface

$$U = \exp(i \int_{\Sigma^+} B^+ - i \int_{\Sigma^-} B^-) \qquad , \qquad \partial \Sigma^+ = \partial \Sigma^- , \tag{3.20}$$

where $\Sigma^\pm$ are two hemispheres in the two sides of the interface. Together they form a closed surface. $U$ acts on the interface lines (3.17) $W \widehat{V}_+^{m^+} \widehat{V}_-^{m^-}$ by a phase of $e^{-2\pi i (m^+ + m^-)/N}$. (As a check, this phase is invariant under the fusion with the trivial operator $a\widehat{V}_+^{-1}\widehat{V}_- = 1$.)

The original $\mathbb{Z}_N$ one-form symmetry acted faithfully on $\mathcal{T}$. This means that there are lines $W$ with all possible $\mathbb{Z}_N$ charges. Therefore, for every value of $m^\pm$ we can find a line $W$ satisfying (3.17). After gauging this $\mathbb{Z}_N$ symmetry, the emergent $\mathbb{Z}_N$ symmetry acts with charge $-(m^+ + m^-)$. We see that it acts faithfully in the resulting TQFT. This means that this emergent $\mathbb{Z}_N$ one-form symmetry is completely broken on the interface.

There is also an emergent dual $\mathbb{Z}_N$ zero-form symmetry on the interface generated by $\exp(i \oint B|)$. All these emergent symmetries have the same origin and gauging them with appropriate counterterms recovers the original system.

We conclude that the 4d-3d system has an emergent $\mathbb{Z}_N$ one-form symmetry, which acts faithfully on the interface; i.e. it is spontaneously broken.

**Effective 3d theory.** Next, we imitate what we did with a boundary and construct an effective interface theory by moding out by the bulk lines

$$V_\pm = (\widehat{V}_\pm)^{K^\pm} = \exp(iK^\pm \oint A^\pm)| . \tag{3.21}$$

Step 2 The bulk lines are trivial in all correlation functions in 3d. We identify them with the trivial lines and therefore, the interface lines $\widetilde{W}$ are identified as

$$\widetilde{W} \sim \widetilde{W} V_- \psi^{K^- p^-/L^-} \sim \widetilde{W} V_+ \psi^{K^+ p^+/L^+} = \widetilde{W} a^{K^+} (\widehat{V}_-)^{K^-} \psi^{K^+ p^+/L^+} , \tag{3.22}$$

where we used the result that $V_\pm$ has interger spin for even $K^\pm p^\pm/L^\pm$ and half integer spin for odd $K^\pm p^\pm/L^\pm$ (see Appendix E).

Step 3 A line at the fixed point of the identification using $a^K$ with $K = \mathrm{lcm}(K^+, K^-)$ can form junctions by emanating two bulk lines $V_+^{K/K^+}$ and $V_-^{K/K^-}$ at the same point. These junctions become genuine line operators if the bulk lines are taken to be trivial.

As an example we consider $\mathcal{T} = (\mathcal{Z}_N)_{-pN}$ defined in (2.14). After gauging all the lines on the interface are generated by $b_+$ and $b_-$

$$b_\pm = \exp(i \oint A^\pm - ip^\pm \oint x). \tag{3.23}$$

We are interested in the expectation value of a knot on the interface

$$K[\{C_i\}, \{C_i'\}] = \exp\left(i \sum_i \oint_{C_i} (A^+ - p^+ x) + i \sum_i \oint_{C_i'} (A^- - p^- x)\right). \tag{3.24}$$

Since the path integral is quadratic it can be evaluated easily (see Appendix E for similar calculations)

$$\langle K[\{C_i\}, \{C_i'\}] \rangle = \exp\left(\frac{2\pi i p^+}{N} \sum_{i<j} \ell(C_i, C_j)\right) \exp\left(-\frac{2\pi i p^-}{N} \sum_{i<j} \ell(C_i', C_j')\right), \tag{3.25}$$

where $\ell(C_i, C_j)$ is the linking number between $C_i$ and $C_j$. Here the result arises from contractions of $\langle A^+ A^+ \rangle$ and $\langle A^- A^- \rangle$. Since $(b_\pm)^{K^\pm}$ is identified with the bulk line $V_\pm = \exp(iK^\pm \oint A^\pm)$, the effective interface theory is $\mathcal{A}^{K^+, -p^+/L^+} \otimes \mathcal{A}^{K^-, p^-/L^-}$.

Using the canonical duality (2.27), the effective interface theory for a general 3d TQFT $\mathcal{T}$ is

$$\frac{\mathcal{T} \otimes \mathcal{A}^{K^+, -p^+/L^+} \otimes \mathcal{A}^{K^-, p^-/L^-}}{\mathbb{Z}_N} = \frac{\mathcal{T}/\mathbb{Z}_L \otimes \mathcal{A}^{K^+, -p^+/L^+} \otimes \mathcal{A}^{K^-, p^-/L^-}}{\mathbb{Z}_K}, \tag{3.26}$$

where the quotient in the first presentation means gauging the diagonal anomaly free $\mathbb{Z}_N$ one-form symmetry generated by $ab^-(b^+)^{-1} = a \exp(i \oint (px - A^+ + A^-))$.

The two minimal Abelian TQFTs $\mathcal{A}^{K^+, -p^+/L^+}$ and $\mathcal{A}^{K^-, p^-/L^-}$ can be interpreted as the 3d TQFTs that the bulk theory provides to cancel the anomaly. The sign difference in the labels comes from the different orientations of the bulk relative to the interface.

It should also be added that when we performed the quotient of the full 4d-3d system by the two bulk theories to find an effective 3d theory, we modded out by the bulk operators. This means that the effective theory captures the correlation functions of interface lines, but does not capture the correlation functions of the bulk lines and the bulk surfaces.

Let us determine the one-form global symmetry of the effective theory. Since we have modded out by some bulk lines, it is different than the $\mathbb{Z}_N$ that acts on all possible lines in the interface.

Clearly, we should focus on the surface operator $U$ that pierces the interface (3.20). In general, it has nontrivial correlation functions with the lines in the bulk. Hence, its intersection with the interface $\partial \Sigma^+ = \partial \Sigma^-$ does not represent a genuine line operator on the interface. Since it is not included as a line operator in our effective theory, the effective theory does not have the full $\mathbb{Z}_N$ symmetry.

However, the surface operator

$$U^{\widetilde{L}}, \qquad \text{with } \widetilde{L} = \text{lcm}(L^+, L^-) \tag{3.27}$$

has trivial correlation functions with all the bulk lines and therefore we expect that it corresponds to a line operator on the interface. Indeed, it is

$$U^{\widetilde{L}} = \left(\widehat{V}_+^{r_+}\right)^{-\widetilde{L}/L^+} \left(\widehat{V}_-^{r_-}\right)^{\widetilde{L}/L^-}, \qquad r_\pm p^\pm = L^\pm \bmod N. \tag{3.28}$$

This line generates a $\mathbb{Z}_{N/\widetilde{L}}$ subgroup of the emergent $\mathbb{Z}_N$ one-form symmetry of the full 4d-3d system.

The one-form global symmetry of the effective theory can also be obtained from (3.26). First, using the $\mathbb{Z}_K$ quotient we can express the symmetry lines as the lines in the minimal Abelian theories. Since $r_\pm p^\pm / L^\pm = 1 \mod K^\pm$, we can choose the generating line of the minimal theories $\mathcal{A}^{K^\pm, \mp p^\pm / L^\pm}$ to be $(\widehat{V}_\pm)^{r_\pm}$. Then the lines in the effective interface theory (3.26) originating from the minimal theories are

$$(\widehat{V}_+^{r_+})^{m^+} (\widehat{V}_-^{r_-})^{m^-}, \qquad m^+ L^+ + m^- L^- = 0 \mod N, \tag{3.29}$$

with $m^\pm \sim m^\pm + K^\pm$. The condition only has solutions $(m^+, m^-) = n(\widetilde{L}/L^+, -\widetilde{L}/L^-)$ with integer $n$ and hence the line (3.28) generates all the interface lines originating from the minimal theories. This means that the $\mathbb{Z}_{N/\widetilde{L}}$ one-form symmetry is the largest symmetry of the effective interface theory (3.26) generated by the lines from the minimal theories.

Another way to understand this global $\mathbb{Z}_{N/\widetilde{L}}$ one-form symmetry of the effective theory is the following. The full 4d-3d system realizes a spontaneously broken $\mathbb{Z}_N$ symmetry, which acts faithfully. In the bulk this symmetry is spontaneously broken to $\mathbb{Z}_{K^\pm}$, so the bulk modes realize $\mathbb{Z}_{L^\pm}$. Together, the two bulk half-spaces realize $\mathbb{Z}_{\widetilde{L}} = \mathbb{Z}_{L^+} \cup \mathbb{Z}_{L^-}$. Therefore, the effective interface theory, obtained as the quotient by the bulk modes realizes $\mathbb{Z}_{N/\widetilde{L}}$. Equivalently, the unbroken global one-form symmetries in the two bulks are $\mathbb{Z}_{K^\pm}$ and hence $\mathbb{Z}_{\gcd(K^+, K^-)} = \mathbb{Z}_{K^+} \cap \mathbb{Z}_{K^-}$ is unbroken throughout the two bulks. We know that the full $\mathbb{Z}_N$ symmetry is broken in the interface. Therefore, the quotient theory should realize the symmetry $\mathbb{Z}_{\gcd(K^+, K^-)} = \mathbb{Z}_{N/\widetilde{L}}$.

# 4 $SU(N)$ and $PSU(N)$ gauge theory in 4d

## 4.1 $SU(N)$ gauge theory, walls and interfaces

We begin by reviewing the dynamics of 4d pure $SU(N)$ gauge theory and its domain walls and interfaces following [1, 2, 42]. The action of the theory is

$$S = -\frac{1}{4g^2} \int \mathrm{Tr}(F \wedge *F) + \frac{\theta}{8\pi^2} \int \mathrm{Tr}(F \wedge F), \tag{4.1}$$

where the parameter $\theta$ is identified periodically $\theta \sim \theta + 2\pi$.

This system has a $\mathbb{Z}_N$ one-form global symmetry, which we will refer to as electric. It is generated by a surface operator

$$\mathbb{U}_{\mathbb{E}} = \exp(i \oint C), \tag{4.2}$$

where $C$ depends on the dynamical gauge fields. As expected of a charge operator, the correlation functions of the surface operator $\mathbb{U}_{\mathbb{E}}$ are topological [1]. The charged objects are Wilson lines in representations of $SU(N)$ and the $\mathbb{Z}_N$ charge is determined by the action of the center of the gauge group on the representation. We will denote the Wilson line in the fundamental representation by $\mathcal{W}$.

In addition to the Wilson lines and the charges $\mathbb{U}_{\mathbb{E}}^r = \exp(ir \oint C)$, the system also includes open versions of the charges

$$T(\gamma) \exp(i \int_\Sigma C), \qquad \gamma = \partial \Sigma, \tag{4.3}$$

where $T$ is the 't Hooft operator. In the $SU(N)$ theory it is not a genuine line operator and needs to be attached to an open surface operator. The 't Hooft operator is the worldline of a monopole, which is defined by being surrounded by a two-sphere with a nontrivial $PSU(N)$ bundle on it. The $SU(N)$ theory does not have such objects. They have to be attached to

strings. (This is like the Dirac string of a magnetic monopole, except that it is detectable by Wilson lines, and hence it is physical.) The surface in (4.3) can be interpreted as the worldsheet of this string. This allows us to interpret the $\mathbb{Z}_N$ charge operator $\mathbb{U}_{\mathbb{E}}^r = \exp(ir \oint C)$ as a closed worldsheet of such strings.

It is natural to couple the global $\mathbb{Z}_N$ symmetry to background gauge fields $\mathcal{B}_C$. Then, since the Wilson lines are charged under the symmetry, they take the form

$$\mathcal{W}(\gamma) e^{\frac{2\pi i}{N} \int_\Sigma \mathcal{B}_C} \qquad , \qquad \gamma = \partial \Sigma \, . \tag{4.4}$$

One way to think about the classical background $\mathcal{B}_C$ is that instead of summing over $SU(N)$ bundles in the path integral, we sum over $PSU(N)$ bundles $\mathcal{E}$ with fixed second Stiefel-Whitney classes $w_2(\mathcal{E}) = \mathcal{B}_C \in H(\mathcal{M}_4, \mathbb{Z}_N)$.

Another consequence of the background field is that we can add to the action the counterterm

$$2\pi \frac{p}{2N} \int_{\mathcal{M}_4} \mathcal{P}(\mathcal{B}_C) \, . \tag{4.5}$$

In the presence of this term the $\theta$ periodicity is as in (1.1)

$$(\theta, p) \sim (\theta + 2\pi, p + N - 1) \, . \tag{4.6}$$

This lack of $2\pi$ periodicity in $\theta$ has another consequence. Because of the Witten effect [43] the open surface operators (4.3) are not invariant under $\theta \to \theta + 2\pi$. They transform as

$$T(\gamma) \exp(i \int_\Sigma C) \to \mathcal{W}(\gamma) T(\gamma) \exp(i \int_\Sigma C) \, . \tag{4.7}$$

This fact will be important below.

So far we have discussed the kinematics of the $SU(N)$ theory. Now we turn to the dynamics. At low energies the $SU(N)$ theory has a gap and it confines. This means that the $\mathbb{Z}_N$ one-form symmetry is unbroken and the charged Wilson lines (those in representations that transform nontrivially under the $\mathbb{Z}_N$ center) have an area law. Correspondingly, these Wilson lines vanish at long distances. As a result, the low-energy theory is trivial. It does not even have a TQFT. In the low-energy theory the Wilson lines $\mathcal{W}^r$ vanish and the charges $\mathbb{U}_{\mathbb{E}}^r$ are equal to one.

The dynamical objects of the system have electric and magnetic charges that are $N$ times the basic units of the Wilson line $\mathcal{W}$ and the 't Hooft operator (with its attached surface (4.3)). Confinement means that some dynamical monopoles or dyons condense. But these are different dyons at $\theta$ and at $\theta + 2\pi$. Because of the Witten effect, their electric charges differ by $N$ units. This means that if we have confinement at $\theta$, we have oblique confinement at $\theta + 2\pi$. And more generally, we have different kinds of oblique confinement at these two values of $\theta$.

At $\theta \in \pi\mathbb{Z}$, the $SU(N)$ gauge theory has a time-reversal symmetry. It is unbroken at $\theta \in 2\pi\mathbb{Z}$. At $\theta \in 2\pi\mathbb{Z} + \pi$, the theory is argued to have two degenerate vacua associated with the spontaneous symmetry breaking of the time reversal symmetry. Since the action of time reversal at these points involve a shift of $\theta$ by a multiple of $2\pi$, the two vacua have different kinds of oblique confinement.

Let us discuss the domain walls between these two vacua. Since they have different kinds of oblique confinement in the two sides, one dyon condenses in one side and another dyon condenses in the other side. Therefore, no dyon condenses on the wall and correspondingly, the theory is not confining there. This means that the electric $\mathbb{Z}_N$ one-form symmetry is spontaneously broken on the domain wall and the fundamental Wilson loops are physical observables in the low-energy theory.

It was argued in [2] that the wall supports a nontrivial TQFT, $SU(N)_1$. This TQFT has a $\mathbb{Z}_N$ one-form symmetry with an anomaly $p = N - 1$, which accounts to the different anomaly

inflow from the two sides of the wall. Note that this is the minimal TQFT with these properties $\mathcal{A}^{N,N-1}$ and any other TQFT with such properties includes $SU(N)_1$ as a decoupled sector and the rest of the theory is $\mathbb{Z}_N$ invariant.

The $SU(N)$ gauge theory can also have interfaces that interpolate between $\theta_0$ and $\theta_0+2\pi k$ for some integer $k$. The anomaly inflow requires the interfaces to support theories with a $\mathbb{Z}_N$ one-form symmetry of anomaly $p = k(N-1)$ mod $2N$. This does not uniquely specify the theories on the interfaces. However, when $\theta$ varies smoothly, the interface theory is uniquely determined by the microscopic theory and the profile of the $\theta$-parameter. This is to be contrasted with sharp interfaces when $\theta$ is discontinuous. When $\theta$ varies smoothly and slowly with $|\nabla\theta| \ll \Lambda$, where $\Lambda$ is the dynamical scale of the theory, there are $k$ domain walls where $\theta$ crosses an odd multiple of $\pi$. Each domain wall supports an $SU(N)_1$ TQFT. When $\theta$ varies smoothly and more rapidly with $|\nabla\theta| \gg \Lambda$, the interface theory $SU(N)_1^{\otimes k}$ is argued to undergo a transition to $SU(N)_k$ theory [2,3]. This can be understood as the Chern-Simons term induced by by the $\theta$-term in the bulk.

However, it is possible that the strong dynamics changes the interface theory at low-energy. One logical possibility is that the dynamics Higgses $SU(N)$ using scalar fields in the adjoint representation. This preserves the $\mathbb{Z}_N$ one-form symmetry and the anomaly. The maximum possible Higgsing with one adjoint scalar is to the Cartan torus $U(1)^{N-1}$, where the $U(1)^{N-1}$ gauge fields $a^I$, $I = 1, \cdots, N-1$ are embedded in the $SU(N)$ gauge field $a$ through

$$a = a^I H^I, \quad (H^I)_{ij} = \mathrm{diag}(\underbrace{0,\cdots,0}_{I-1}, 1, -1, \underbrace{0,\cdots,0}_{N-I-1}). \tag{4.8}$$

In terms of these fields the $SU(N)_k$ theory becomes a $U(1)^{N-1}$ Chern-Simons theory

$$\frac{k}{4\pi}\mathrm{Tr}\left(ada - \frac{2i}{3}a^3\right) \;\to\; \frac{k}{4\pi}(K_{\mathrm{Cartan}})_{IJ}a^I da^J, \tag{4.9}$$

where $K_{\mathrm{Cartan}}$ is the Cartan matrix of $SU(N)$

$$(K_{\mathrm{Cartan}})_{IJ} = \mathrm{Tr}(H^I H^J) = 2\delta_{I,J} - \delta_{I,J+1} - \delta_{I+1,J}. \tag{4.10}$$

For $k = 1$ this Abelian TQFT is the same as $SU(N)_1$, so this possibility is the same as the previous suggestion.

We can further Higgs $SU(N)$ all the way down to its $\mathbb{Z}_N$ center. In order to identify the TQFT of this $\mathbb{Z}_N$ gauge theory, we use a presentation of $SU(N)_k$ based on $U(N) \times U(1)$ gauge fields $b$ and $y$ [6]

$$\frac{k}{4\pi}\mathrm{Tr}\left(bdb - \frac{2i}{3}b^3\right) - \frac{k}{4\pi}(\mathrm{Tr}\,b)d(\mathrm{Tr}\,b) + \frac{1}{2\pi}y\,d(\mathrm{Tr}\,b), \tag{4.11}$$

where the $U(1)$ field $y$ constrains $b$ to be a $SU(N)$ gauge field. The $\mathbb{Z}_N$ gauge field $x$ is embedded in $U(N)$ through $b = x\mathbb{I}$. After Higgsing, the $SU(N)_k$ theory becomes a $\mathbb{Z}_N$ gauge theory $(\mathcal{Z}_N)_{-kN(N-1)} = (\mathcal{Z}_N)_{-pN}$. Alternatively, the precise $\mathbb{Z}_N$ gauge theory can be determined by matching the anomalies.

In conclusion, without a more detailed dynamical analysis we cannot uniquely determine the TQFT on the interface, so we will denote it by $\mathcal{T}_k$. The simplest case $\mathcal{T}_1$ was argued to be the minimal allowed theory, $SU(N)_1$. But for higher values of $k$ there isn't a preferred choice and we presented several options, e.g. $SU(N)_k$ and $(\mathcal{Z}_N)_{-kN(N-1)} = (\mathcal{Z}_N)_{-pN}$. However, using the analysis in the previous sections, we can proceed without knowing exactly what $\mathcal{T}_k$ is.

Let us analyze the interface theory $\mathcal{T}_k$ in more detail. The theory has a $\mathbb{Z}_N$ one-form symmetry of anomaly $k(N-1)$, which means that the symmetry lines are anyons with a braiding phase of $e^{-2\pi i k(N-1)/N}$. These symmetry lines can be thought of as bulk charge operators generated by $\mathbb{U}_{\mathbb{E}}$ that pierce the interface. To see that, recall that because of confinement, the

shape of $\mathbb{U}_\mathbb{E}$ in the bulk is not important (a closed surface on each side equals to one) and therefore, $\mathbb{U}_\mathbb{E}$, which pierces the interface is effectively a line operator on the interface. Also, $\mathbb{U}_\mathbb{E}$ can be interpreted as the worldsheet of a string constructed by gluing two 't Hooft lines from the two sides at the interface. So we can view $\mathbb{U}_\mathbb{E}$ as associated with two 't Hooft lines, $T$ on one side of the interface and $T^{-1}$ on the other side. Then, because of the Witten effect [43], the electric charges of these two 't Hooft lines differ by $k$ and therefore $\mathbb{U}_\mathbb{E}$ that pierces the interface appears as a Wilson line with electric charge $k$. More precisely, it is the generator of the $\mathbb{Z}_N$ one-form global symmetry on the interface. For example, if the theory on the interface $\mathcal{T}_k$ is $SU(N)_k$, it is a Wilson line in a $k$ index symmetric representation of $SU(N)$.

The fact that $\mathbb{U}_\mathbb{E}$ leads to a Wilson line on the interface shows that not only are the probe quarks on the interface liberated (because there is no confinement there), they are also anyons!

## 4.2 $PSU(N)$ gauge theory

The $PSU(N)$ gauge theory differs from the $SU(N)$ gauge theory in the global form of the gauge group. It can be constructed by gauging the electric $\mathbb{Z}_N$ one-form symmetry in the $SU(N)$ gauge theory, i.e. by making the classical background field $\mathcal{B}_\mathcal{C}$ dynamical (and dropping the subscript $\mathcal{C}$). Summing over $\mathcal{B}$ means that we sum over all $PSU(N)$ bundles $\mathcal{E}$. Now, the choice of the counterterm (4.5) is more significant than in the $SU(N)$ theory and the value of $p$ affects the set of observables.

Let us discuss the operators in the theory. Since now $\mathcal{B}$ is dynamical, the Wilson loop (4.4) is no longer a genuine line operator; it depends on the surface $\Sigma$. We can consider a closed surface operator

$$\mathbb{U}_\mathbb{M} = \exp\left(\frac{2\pi i}{N} \oint w_2^{PSU(N)}\right) = \exp\left(\frac{2\pi i}{N} \oint \mathcal{B}\right), \tag{4.12}$$

where $w_2^{PSU(N)}$ is the abbreviation for $w_2(\mathcal{E})$ (with $\mathcal{E}$ the $PSU(N)$ bundle). It is the generator of a new emergent $\mathbb{Z}_N$ one-form symmetry, which we will refer to as magnetic.

The original Wilson line is an open version of $\mathbb{U}_\mathbb{M}$. And just as the surface in this Wilson line can be interpreted as the worldsheet of an electric (confining) string, the closed surface operator $\mathbb{U}_\mathbb{M}$ can be interpreted as a closed worldsheet of such a string. (Note that in the $PSU(N)$ theory this string worldsheet is an operator in the theory.)

For $p = 0$ the 't Hooft line $T$ is a genuine line operator and we do not need to write $C$ of (4.3). It is charged under the magnetic symmetry (4.12). Other dyonic operators of the form $T\mathcal{W}^r$ need an attached surface and they are not genuine line operators (unless $r = 0 \mod N$).

We would like to find the line operators when $p$ is nonzero. We simplify the discussion by considering the theory on a spin manifold such that the periodicity of $p$ is $p \sim p + N$.[26] We first keep $p = 0$ and extend the range of $\theta \sim \theta + 2\pi N$. Clearly, $T$ remains a genuine line operator as we change $\theta$. But because of the Witten effect it acquires electric charge $-k$ as $\theta$ is shifted by $-2\pi k$. Then we restore the original $\theta$ and have nonzero $p = k(N-1)$. This means that the basic line operator has electric charge $p$, i.e. it is [10]

$$\widehat{T}(\gamma) = T(\gamma)\mathcal{W}(\gamma)^p . \tag{4.13}$$

Note that this is a genuine line operator, which does not need a surface.

---

[26]On an orientable non-spin manifold, the change $p \rightarrow p + N$ (with even $N$) produces the coupling $\pi \int w_2(\mathcal{M}_4) \cup \mathcal{B}$ (where $w_2(\mathcal{M}_4)$ is the second Stiefel-Whitney class of the 4d manifold $\mathcal{M}_4$) that is equivalent to turning on classical background field $\widetilde{\mathcal{B}}_\mathcal{C} = N w_2(\mathcal{M}_4)/2$ for the magnetic $\mathbb{Z}_N$ one-form symmetry generated by $\exp(\frac{2\pi i}{N} \oint \mathcal{B})$. Thus it changes the statistics of the basic 't Hooft line from a boson to a fermion and vice versa [39, 44, 45]. This does not modify the $PSU(N)$ bundle but instead gives additional weights in the path integral.

Table 5: The dictionary between the operators in the microscopic $PSU(N)$ gauge theory and the operators in the macroscopic $\mathbb{Z}_N$ two-form gauge theory. The line operator in the second row is the minimal line that obeys a perimeter law. It is identified with the genuine line operator in the low-energy theory (and hence we suppress the $B$ dependent term). Here we use a continuous notation for the low-energy TQFT, which is reviewed in appendix E.

| Microscopic $PSU(N)$ gauge theory | Low energy $\mathbb{Z}_N$ two-form gauge theory |
|---|---|
| $\widehat{T}^r = (T\mathcal{W}^p)^r$ | 0 for $r \neq 0 \bmod K$ |
| $\widehat{T}^K = T^K \mathcal{W}^{pK}$ | $\exp\left(iK \oint A + ipK \int B\right) \sim \exp\left(iK \oint A\right)$ |
| $\mathbb{U}_\mathbb{M} = \exp\left(\frac{2\pi i}{N} \oint w_2^{PSU(N)}\right)$ | $\exp\left(i \oint B\right)$ |

Another way to understand the lines (4.13) is to write them as $T\mathcal{W}^p \exp\left(i \int_\Sigma (C + \frac{2\pi p}{N}\mathcal{B})\right)$, where $C$ comes from $T$ (4.3) and $\frac{2\pi p}{N}\mathcal{B}$ from $\mathcal{W}$ (4.4). In the $PSU(N)$ theory with $p$ the term in the exponent vanishes and hence this operator is independent of $\Sigma$.

Now, let us consider the dynamics. In the $SU(N)$ theory the dyons that condense at $\theta = 2\pi k$ have the quantum numbers of $T^N \mathcal{W}^{kN}$. (Note that these dyons exist as dynamical objects regardless of the global structure of the gauge group. The global part of the group and the value of $p$ determine the line operators in the theory.)

Let us focus on $\theta = 0$ with arbitrary $p$. The genuine line operators in the theory are powers of $\widehat{T}$ (4.13). Some of them have area law because of the condensation and hence they vanish at low energies. Only the lines that are generated by

$$\widehat{T}^K = T^K \mathcal{W}^{pK} \quad , \quad L = \gcd(N, p) \quad , \quad K = \frac{N}{L} , \tag{4.14}$$

are aligned with the condensed dyons and hence they have a perimeter law. These are the only nontrivial line operators in the low-energy theory.

It is clear that the magnetic $\mathbb{Z}_N$ one-form symmetry is spontaneously broken to $\mathbb{Z}_K$ and the broken elements are realized at low-energy by a $\mathbb{Z}_L$ gauge theory [10]. The operators in this $\mathbb{Z}_L$ gauge theory are generated by the basic $\mathbb{Z}_L$ Wilson line (4.14) (which is not to be confused with the microscopic $PSU(N)$ Wilson line) and its dual surface operator, which is the microscopic operator $\mathbb{U}_\mathbb{M}$.[27]

In conclusion, the low-energy manifestation of this spontaneous symmetry breaking of the magnetic $\mathbb{Z}_N$ one-form symmetry is the theory (3.2). And the relation between the microscopic operators in the $PSU(N)$ gauge theory and the low-energy theory is summarized in Table 5.

### 4.3 Interfaces in $PSU(N)$ gauge theory

Here we study an interface in the $PSU(N)$ theory. We let it interpolate smoothly between $\theta = 0$ and $\theta = 2\pi k$. As above, we can approximate it at low energies with constant $\theta = 0$ and $p$ changing from $p^+$ to $p^-$. This is the setup we considered in the $SU(N)$ theory above, and now we simply gauge the electric $\mathbb{Z}_N$ one-form symmetry in that theory.

We use the definitions (3.16)

$$L^\pm = \gcd(N, p^\pm), \quad L = \gcd(L^+, L^-), \quad K^\pm = N/L^\pm, \quad K = N/L . \tag{4.15}$$

The low-energy dynamics of the $PSU(N)$ theory in the two sides are approximated by the $\mathbb{Z}_N$ two-form gauge theories with parameters $p^\pm$, which are equivalent to $\mathbb{Z}_{L^\pm}$ gauge theories. They describe the spontaneous breaking of the magnetic $\mathbb{Z}_N$ one-form global symmetry to

---

[27]On a nonspin manifold this $\mathbb{Z}_L$ gauge theory could be twisted, as in Appendix E.

$\mathbb{Z}_{K^{\pm}}$. Note that unlike the $SU(N)$ theory, where the two sides of the interface differed only by a counterterm for background fields, here the two sides are dynamically different.

The TQFT in the bulk and on the interface is as in Section 3.3, so we will not repeat its analysis in detail, except to summarize the main points.

We have already said that in the bulk the magnetic $\mathbb{Z}_N$ one-form symmetry is spontaneously broken to $\mathbb{Z}_{K^{\pm}}$. On the interface, since the confined line operators in the bulk become liberated, the magnetic $\mathbb{Z}_N$ one-form symmetry, generated by the surface operators piercing the interface, is completely broken. Equivalently, we have argued above that in the $SU(N)$ theory no monopole condenses on the wall and the dynamics is weakly coupled there. Therefore, the $\mathbb{Z}_N$ one-form symmetry of the $PSU(N)$ theory should also be spontaneously broken there.

When $\theta$ varies smoothly and rapidly, the interface in the $SU(N)$ gauge theory supports a TQFT $\mathcal{T}_k$. The effective interface theory on the corresponding $PSU(N)$ interface is found easily using the results in Section 3.3. When $L^+ = L^- = 1$ the theory on the interface is

$$
\frac{\mathcal{T}_k \otimes \mathcal{A}^{N,-p^+} \otimes \mathcal{A}^{N,p^-}}{\mathbb{Z}_N} \ . \tag{4.16}
$$

As in Section 3.3, we can interpret the two minimal theories in the numerator as produced by the bulk in the two sides, such that we can gauge an anomaly free $\mathbb{Z}_N$ one-form global symmetry.

For generic $L^{\pm}$ the interface couples to the $\mathbb{Z}_{L^{\pm}}$ gauge theory in the bulk and it is meaningless to ask what the theory on the interface is. Yet, we can identify an effective interface theory. It is

$$
\frac{\mathcal{T}_k/\mathbb{Z}_L \otimes \mathcal{A}^{N/L^+,-p^+/L^+} \otimes \mathcal{A}^{N/L^-,p^-/L^-}}{\mathbb{Z}_K} = \frac{\mathcal{T}_k \otimes \mathcal{A}^{N/L^+,-p^+/L^+} \otimes \mathcal{A}^{N/L^-,p^-/L^-}}{\mathbb{Z}_N} \ . \tag{4.17}
$$

As an example, we argued above that the interface in the $SU(N)$ theory between $\theta = 0$ and $\theta = 2\pi$ with $p^+ = p^- = 0$ supports an $SU(N)_1$ theory. This corresponds to $\theta = 0$ with $p^+ = 0$ and $p^- = 1-N$, and thus $L^+ = N, L^- = 1$. The effective interface theory on the corresponding $PSU(N)$ interface is trivial, since $\mathcal{A}^{N,1-N} = SU(N)_{-1}$.

# Acknowledgements

We thank Maissam Barkeshli, Meng Cheng, Clay Córdova, Dan Freed, Davide Gaiotto, Jaume Gomis, Zohar Komargodski, Kantaro Ohmori, Shu-Heng Shao, Zhenghan Wang, and Edward Witten for useful discussions. The work of P.-S.H. was supported by the Department of Physics at Princeton University, the Institute for Advanced Study, and the U.S. Department of Energy, Office of Science, Office of High Energy Physics, under Award Number DE-SC0011632, and by the Simons Foundation through the Simons Investigator Award. The work of H.T.L. is supported by a Croucher Scholarship for Doctoral Study, a Centennial Fellowship from Princeton University and the Institute for Advanced Study. The work of N.S. is supported in part by DOE grant DE-SC0009988.

# A   Definitions of Abelian anyons

In this Appendix we will review some properties of Abelian anyons. There are three equivalent definitions of Abelian anyons. An anyon $a$ in a $3d$ TQFT is called Abelian when

(1) $a$ obeys group-law fusion, $aa^s = a^{s+1}$ for integers $s$ with $a^0 = 1$. In particular, since the number of lines in a consistent 3$d$ TQFT is finite, there exists an integer $m$ such that $a^m = 1$.

(2) $a$ obeys Abelian fusion rules. For any line $W$ in the 3$d$ TQFT, the fusion product $aW$ only contains one line.

(3) the quantum dimension of $a$ is one.

First, the definition (1) implies (3). The group-law fusion $a^m = aa \cdots a = 1$ implies $d_a^m = 1$ for the quantum dimension $d_a$ of $a$. Since $d_a$ must be a positive real number in any unitary 3$d$ TQFT, we conclude $d_a = 1$.

The definition (2) implies (1) by specializing $W = a, aa, \cdots$ and defining the unique line appears in the fusion of $n$ line $a$ to be $a^n$.

Now we will show the definition (3) implies (2) by contradiction. Suppose there exists a line $x$ that fuses with $a$ into at least two lines that we denote by $y, z$:

$$a \cdot x = y + z + \cdots . \tag{A.1}$$

This implies

$$\overline{a} \cdot y = x + \cdots , \tag{A.2}$$

where $\overline{a}$ denotes the antiparticle of $a$, *i.e.* $a \cdot \overline{a} = 1 + \cdots$. The quantum dimensions in the fusion $u \cdot v = \sum_i w_i$ satisfy $d_u d_v = \sum_i d_{w_i}$ [22], and thus

$$d_a d_x = d_y + d_z + \cdots , \quad d_{\overline{a}} d_y = d_x + \cdots \quad \Rightarrow \quad d_a d_{\overline{a}} d_x \geq d_x + d_{\overline{a}} d_z > d_x , \tag{A.3}$$

where the last two inequalities used the property that the quantum dimensions are real and positive, and in particular the last inequality comes from the existence of the second anyon $z$ in the fusion (A.1). Since $\overline{a}$ and $a$ have the same quantum dimension, by definition (3) $d_a = d_{\overline{a}} = 1$. Thus the last equation in (A.3) leads to a contradiction. Therefore, any line $x$ must fuse with $a$ into only one line. We conclude that (3) implies (2), and since we have already shown that (1) implies (3), this means that (1) implies (2). This completes the proof that the three definitions are equivalent to one another.

# B Jacobi symbols

For any odd prime number $q$, the Legendre symbol is defined as

$$\left( \frac{a}{q} \right) = a^{\frac{q-1}{2}} \bmod q = \begin{cases} 0 & a = 0 \mod q \\ 1 & a = r^2 \bmod q \text{ for some integer } r \\ -1 & \text{otherwise} \end{cases} . \tag{B.1}$$

For any odd integer $b$ with a prime factorization $b = \prod_k q_k^{\alpha_k}$, the Jacobi symbol is the generalization of the Legendre symbol defined as

$$\left( \frac{a}{b} \right) = \prod_k \left( \frac{a}{q_k} \right)^{\alpha_k} . \tag{B.2}$$

The Jacobi symbol obeys the following identities for odd integers $a, b, c$

$$\left( \frac{ab}{c} \right) = \left( \frac{a}{c} \right) \left( \frac{b}{c} \right) , \quad \left( \frac{-1}{c} \right) = (-1)^{(c-1)/2} . \tag{B.3}$$

# C  2d unitary chiral RCFT for Abelian 3d TQFT

In this Appendix we will show that every Abelian 3d TQFT corresponds to a 2d unitary chiral RCFT. Such unitary CFTs are generally not unique for a given TQFT and here we construct one example of them. The unitary RCFT is characterized by an extended chiral algebra of a product of chiral algebras of free compact bosons, free complex fermions, and $SU(N)_1$ Wess-Zumino-Witten models. If the TQFT is a spin theory, then the RCFT is $\mathbb{Z}_2$-graded [9].

Every Abelian TQFT $\mathcal{A}$ can be expressed as an Abelian Chern-Simons theory [28] [27–31] (for a review see *e.g.* [32]). Denote the $U(1)$ gauge fields by $x_0, x_1, \cdots, x_n$ for some integer $n$, and the Chern-Simons action is

$$\frac{k}{4\pi} x_0 dx_0 + \sum_{i=1}^{n} \left( \frac{q_{0i}}{2\pi} x_0 dx_i \right) + \mathcal{L}[x_1, \cdots, x_n] \,, \tag{C.1}$$

where $k, q_{0i}$ are integers, and $\mathcal{L}[x_1, \cdots, x_n]$ denotes Chern-Simons terms independent of the gauge field $x_0$. $k, q_{0i}$ cannot be simultaneously zero for all $i$, since otherwise the theory has a decoupled gapless sector described by the dual photon of $x_0$. If $k = 0$, there exists $q_{0i} \neq 0$ for some $i$, and the redefining $x_i \to x_i + x_0$ produces nonzero $k$. Thus we can assume $k$ is always nonzero without loss of generality. Consider the change of variables from $x_0, x_1, \cdots, x_n$ to $y_0, y_1, \cdots, y_n$

$$x_0 = y_0 - \sum_{i=1}^{n} q_{0i} y_i, \quad x_j = k y_j, \ j = 1, \cdots, n \,. \tag{C.2}$$

The Jacobian is $|k|^n$. The theory $\mathcal{A}$ can thus be expressed as

$$\mathcal{A} = \frac{\mathcal{A}'}{\mathbb{Z}_{|k|}^n} \,, \tag{C.3}$$

where the quotient denotes gauging a one-form symmetry $\mathbb{Z}_{|k|}^n$, and $\mathcal{A}'$ is an Abelian Chern-Simons theory with $U(1)$ gauge fields $y_0, y_1, \cdots, y_n$. Substituting (C.2) into (C.1), we find the theory $\mathcal{A}'$ has the Chern-Simons action

$$\frac{k}{4\pi} y_0 dy_0 + \widetilde{\mathcal{L}}[y_1, \cdots, y_n] \,, \tag{C.4}$$

where $\widetilde{\mathcal{L}}[y_1, \cdots, y_n]$ denotes Chern-Simons terms independent of $y_0$. Thus $\mathcal{A}' = U(1)_k \otimes \mathcal{A}''$ for another Abelian Chern-Simons theory $\mathcal{A}''$ with gauge fields $y_1, \cdots, y_n$. By iteration, we find the Abelian TQFT $\mathcal{A}$ can be expressed as

$$\mathcal{A} = \widehat{\mathcal{A}}/\mathcal{Z}, \quad \widehat{\mathcal{A}} = \prod_{i=0}^{n} U(1)_{k_i} \,, \tag{C.5}$$

where the quotient denotes gauging a one-form symmetry $\mathcal{Z}$ that is a finite Abelian group, and $k_i$ are non-zero integers.

If all $k_i$ are positive, then the Abelian TQFT $\mathcal{A}$ corresponds to the extended chiral algebra of a product of compact bosons in 2d (the RCFT may be $\mathbb{Z}_2$ graded).

If some of $k_i = -m_i$ is negative, the corresponding $U(1)_{-m_i}$ in $\widehat{\mathcal{A}}$ can be replaced by an $SU(N)$ Chern-Simons theory at level one using the duality [29]

$$U(1)_{-m_i} \quad \longleftrightarrow \quad \begin{cases} SU(4m_i)_1/\mathbb{Z}_2 & \text{even } m_i \\ SU(m_i)_1 \otimes \{1, \psi\} & \text{odd } m_i \end{cases}, \tag{C.6}$$

---

[28]For example, the Chern-Simons theories with gauge group of rank $n$ including $SU(n+1)_1$, $Spin(2n)_1$ and $(E_n)_1$ can be written as $U(1)^n$ Abelian Chern-Simons theories with the coefficient matrix given by the Cartan matrix of the gauge groups.

[29]For the case $m_i$ is odd, the duality (C.6) is the level-rank duality [6].

where for even $m_i$ the theory $U(1)_{-m_i}$ is non-spin, and we omit a trivial TQFT such as $(E_8)_1$ in the duality.

For odd $m_i$ the theory $U(1)_{-m_i}$ is a spin theory. On the right hand side of the duality (C.6) the theory $\{1, \psi\}$ represents the almost trivial TQFT that has only two lines (of integer and half integer spins), and it includes the gravitational Chern-Simons term $-2M_i \mathrm{CS}_g$ for some positive integer $M_i = -m_i$ mod 8. The almost trivial TQFT corresponds to $M_i$ free complex fermions in 2d.

Thus the theory $\widehat{\mathcal{A}}$ corresponds to the 2d unitary chiral RCFT ($\mathbb{Z}_2$ graded if some $k_i$ is odd) given by the product of free compact bosons, free complex fermions, and $SU(N_i)$ Wess-Zumino-Witten models at level one with $N_i$ given in (C.6) (or its extended chiral algebra when $k_i = -m_i$ is even and negative). The Abelian TQFT $\mathcal{A}$ then corresponds to the 2d unitary chiral RCFT given by the extended chiral algebra (C.5) of the 2d unitary chiral RCFT of $\widehat{\mathcal{A}}$.

# D   Gauging a general anomaly free subgroup

In order to simplify the discussion we will assume in this appendix that all the TQFTs are spin TQFTs.

A theory $\mathcal{T}$ with a $\mathbb{Z}_N$ one-form symmetry with anomaly $p$ can have multiple anomaly free subgroups. One of them is the $\mathbb{Z}_L$ subgroup with $L = \gcd(N, p)$. In this Appendix, we will discuss gauging a larger anomaly free symmetry $\mathbb{Z}_m$, i.e.

$$\mathbb{Z}_L \subset \mathbb{Z}_m \subset \mathbb{Z}_N \ . \tag{D.1}$$

It is anomaly free when $pN/m^2$ is an integer (recall that we discuss spin theories). Gauging this symmetry leads to $\mathcal{T}/\mathbb{Z}_m$, which has a $\mathbb{Z}_{N'}$ one-form symmetry of anomaly $p'$ with

$$N' = \frac{NL}{m^2} \qquad , \qquad p' = \frac{p}{L} \ . \tag{D.2}$$

They satisfy $\gcd(N', p') = 1$. Then we can further apply the generalized gauging operation with respect to this $\mathbb{Z}_{N'}$ one-form symmetry to find

$$\frac{\mathcal{T}/\mathbb{Z}_m \otimes \mathcal{A}^{N', -p'}}{\mathbb{Z}_{N'}} \ . \tag{D.3}$$

The goal of this appendix is to show that this is the same as the answer in (2.35)

$$\frac{\mathcal{T}/\mathbb{Z}_L \otimes \mathcal{A}^{N/L, -p/L}}{\mathbb{Z}_{N/L}} = \frac{\mathcal{T} \otimes \mathcal{A}^{N/L, -p/L}}{\mathbb{Z}_N} \ . \tag{D.4}$$

Note, as a check that for $L = m$ they are trivially the same.

We will use the canonical duality in (2.27) [36]

$$\mathcal{T} \longleftrightarrow \frac{\mathcal{T} \otimes (\mathcal{Z}_N)_{-pN}}{\mathbb{Z}_N} \ . \tag{D.5}$$

The second factor in the numerator can be described by the Lagrangian (2.14)

$$\int \left( -\frac{pN}{4\pi} x dx + \frac{N}{2\pi} x dy \right). \tag{D.6}$$

Its lines are generated by $b$ and $c$ (2.15)

$$b = \exp(i \oint y), \quad c = \exp(ip \oint x - i \oint y). \tag{D.7}$$

In this dual description, the $\mathbb{Z}_N$ one-form symmetry is entirely in the $(\mathscr{Z}_N)_{-pN}$ factor and it is generated by $b$.

The duality allows us to only keep tack of the $(\mathscr{Z}_N)_{-pN}$ factor in various procedures (and ignore the TQFT $\mathcal{T}$).

Gauging the anomaly free $\mathbb{Z}_L$ subgroup in $(\mathscr{Z}_N)_{-pN}$ is the same as redefining $x$ as $x' = Lx$ and viewing $x'$ as a $U(1)$ gauge field. This leads to $(\mathscr{Z}_K)_{-p'K}$ with $K = N/L$ and $p' = p/L$. Since $\gcd(K, p') = 1$, the theory $(\mathscr{Z}_K)_{-p'K}$ factorizes (2.26)

$$(\mathscr{Z}_K)_{-p'K} = \mathcal{A}^{K,p'} \otimes \mathcal{A}^{K,-p'}, \tag{D.8}$$

where the first and second minimal theories are generated by $b$ and $c$, respectively.

Then, gauging the anomaly free $\mathbb{Z}_m \subset \mathbb{Z}_N$ (which includes $\mathbb{Z}_L$) in $(\mathscr{Z}_N)_{-pN}$ is equivalent to gauging the anomaly free $\mathbb{Z}_{m/L}$ subgroup generated by $b^{N/m}$ in $(\mathscr{Z}_K)_{-p'K} = \mathcal{A}^{K,p'} \otimes \mathcal{A}^{K,-p'}$. Only the first minimal theory is involved in the gauging, which reduces it to $\mathcal{A}^{N',p'}$ with $N' = K(L/m)^2 = NL/m^2$ and $p' = p/L$.[30] This implies that

$$\frac{\mathcal{T}}{\mathbb{Z}_m} \longleftrightarrow \left(\frac{\mathcal{T} \otimes (\mathscr{Z}_N)_{-pN}}{\mathbb{Z}_N}\right)\Big/\mathbb{Z}_m \longleftrightarrow \frac{\mathcal{T} \otimes \mathcal{A}^{N',p'} \otimes \mathcal{A}^{K,-p'}}{\mathbb{Z}_N}. \tag{D.9}$$

The remaining global symmetry is $\mathbb{Z}_{N'}$ and it is carried by the second factor in the numerator. Applying the generalized operation with respect to this symmetry removes this factor and leads to

$$\frac{\mathcal{T} \otimes \mathcal{A}^{N/L,-p/L}}{\mathbb{Z}_N}. \tag{D.10}$$

We conclude that the final theory (D.10) is the same for any choice of $\mathbb{Z}_m \supset \mathbb{Z}_L$.

# E    Two-form $\mathbb{Z}_N$ gauge theory in 4d

The 4d topological $\mathbb{Z}_N$ two-form gauge theory of a gauge field $\mathcal{B} \in \mathcal{H}^2(\mathcal{M}_4, \mathbb{Z}_N)$

$$S = 2\pi \frac{p}{2N} \int \mathcal{P}(\mathcal{B}), \tag{E.1}$$

has a continuum description [1,4]

$$S = \int \left(\frac{pN}{4\pi} BB + \frac{N}{2\pi} BdA\right), \tag{E.2}$$

where $A$ is a $U(1)$ one-form gauge field and $B$ is a $U(1)$ two-form gauge field. $A$ constrains $B$ to be a $\mathbb{Z}_N$ two-form gauge field $B \to \frac{2\pi}{N}\mathcal{B}$. The theory has a one-form gauge symmetry

$$B \to B - d\lambda, \quad A \to A + p\lambda. \tag{E.3}$$

Under the gauge transformation, the action is shifted by

$$-\int \left(\frac{pN}{4\pi} d\lambda d\lambda + \frac{N}{2\pi} d\lambda dA\right). \tag{E.4}$$

---

[30]More generally, $\mathcal{A}^{M,r}$ with $\gcd(M, r) = 1$ is generated by a line $z$ such that $z^M = 1$ and the spin of $z$ is $\frac{r}{2M}$. When $M = \widehat{M}q^2$ with $\widehat{M}, q \in \mathbb{Z}$, it has a $\mathbb{Z}_q$ anomaly free subgroup generated by $z^{\widehat{M}q}$. (It is anomaly free because the spin of this line is $\frac{r\widehat{M}}{2}$.) The gauged theory $\mathcal{A}^{M,r}/\mathbb{Z}_q$ has $\widehat{M}$ lines generated by $z^q$ (with $(z^q)^{\widehat{M}} = 1$), whose spin is $\frac{r}{2\widehat{M}}$. Therefore, the resulting theory is $\mathcal{A}^{M,r}/\mathbb{Z}_q = \mathcal{A}^{\widehat{M},r}$.

On a closed spin manifolds it is always a multiple of $2\pi$, but on general closed manifolds it is a multiple of $2\pi$ only when $pN$ is even. The parameter $p$ has an identification of $p \sim p + 2N$ on non-spin manifolds and $p \sim p + N$ on spin manifolds.

Define

$$L = \gcd(N, p), \quad K = N/L .$$ 
(E.5)

The theory has $L$ surface operators generated by

$$U = \exp(i \oint B), \quad U^L = 1$$ 
(E.6)

and $L$ genuine lines operators generated by

$$V = \exp(iK \oint_{\partial\Sigma} A + ipK \int_\Sigma B), \quad V^L = 1$$ 
(E.7)

(they are genuine line operators because they do not depend on the surface $\Sigma$). These operators and their correlation functions are identical to the ones in a $\mathbb{Z}_L$ gauge theory, and they realize a $\mathbb{Z}_L = \mathbb{Z}_N/\mathbb{Z}_K$ one-form symmetry. As we will discuss below, depending on $N$ and $p$ this $\mathbb{Z}_L$ gauge theory could be twisted on nonspin manifolds.

This theory can arise as the low-energy approximation of a microscopic theory whose $\mathbb{Z}_N$ one-form symmetry is spontaneously broken to $\mathbb{Z}_K$. Examples of such UV theories are a $PSU(N)$ gauge theory (discussed in Section 4) and the Walker-Wang lattice model [18,19,46].

There are also open surface operators generated by

$$\exp(i \oint_{\partial\Sigma} A + ip \int_\Sigma B).$$ 
(E.8)

They are genuine line operators if the surface dependence is trivial, otherwise, the surface is physical and the operators can only have contact terms. Hence, we will not include them in the list of operators.

Two special cases are particularly interesting. First, for $p = 0$ this theory is the same as an ordinary $\mathbb{Z}_N$ gauge theory. Here $B$ implements the constraint that $A$ is a $\mathbb{Z}_N$ one-form gauge field.

The second special case is $p = N$. On a spin manifold, it is the same as $p = 0$, i.e. it is an ordinary $\mathbb{Z}_N$ gauge theory. On a nonspin manifold, we must have $pN \in 2\mathbb{Z}$ so, $p = N$ can happen only when $N$ is even. Then, the action (E.1) is the same as

$$\pi \int \mathcal{P}(\mathcal{B}) = \left( \pi \int w_2(\mathcal{M}_4) \cup \mathcal{B} \right) \bmod 2\pi ,$$ 
(E.9)

where $w_2(\mathcal{M}_4)$ is the second Stiefel-Whitney class of the manifold. This fact has some interesting consequences. First, it shows that the possible added term (E.9) on nonspin manifolds for even $N$ was already included in our labelling by $p = 0, 1, \cdots, 2N - 1$. Second, it makes it manifest that on spin manifolds we can identify $p \sim p + N$. Finally, it shows that on a nonspin manifold, the theory with even $p = N$, which is an ordinary $\mathbb{Z}_N$ gauge theory on a spin manifold, becomes a $\mathbb{Z}_N$ gauge theory coupled to $w_2(\mathcal{M}_4)$ of the manifold.

In the $\mathbb{Z}_N$ gauge theory, the surface $\oint \mathcal{B}$ is the world volume of a $\mathbb{Z}_N$ magnetic string. It generates the one-form symmetry that acts on the Wilson lines in the $\mathbb{Z}_N$ gauge theory. The coupling (E.9) is thus equivalent to turning on a background gauge field for this one-form symmetry $\widetilde{\mathcal{B}}_C = (N/2)w_2(\mathcal{M}_4) \bmod N$. One consequence of this is that on a non-spin manifold, the basic $\mathbb{Z}_N$ Wilson line, which corresponds to the microscopic line $\oint A$, is attached to the surface $\frac{2\pi}{N} \int \widetilde{\mathcal{B}}_C = \pi \int w_2(\mathcal{M}_4)$. The surface represents an anomaly in the theory along

the line and it implies that if we view this line as the worldline of a probe particle, this particle is a fermion [39, 44]. The conclusion is that the theory with $p = N$ for even $N$ is a (twisted) $\mathbb{Z}_N$ gauge theory with fermionic probe particles.

Another way to see this is as follows. $w_2(\mathcal{M}_4)$ of a manifold is the obstruction to lifting the $SO(4)$ tangent bundle to an $Spin(4)$ bundle. Thus the background $\widetilde{\mathcal{B}}_{\mathcal{C}} = (N/2)w_2(\mathcal{M}_4)$ modifies the symmetry to be

$$\frac{\mathbb{Z}_N^{\text{gauge}} \times Spin(4)}{\mathbb{Z}_2} . \tag{E.10}$$

The quotient identifies $\mathbb{Z}_2 \subset \mathbb{Z}_N^{\text{gauge}}$ with the $\mathbb{Z}_2$ fermion parity symmetry $(-1)^F$ of the Lorentz symmetry. Thus the $\mathbb{Z}_N$ Wilson lines in the odd-charge representations also transform under the fermion parity, and they represent fermionic probe particles.

Let us examine in more detail the path integral of the $\mathbb{Z}_N$ gauge theory coupled to fixed $w_2(\mathcal{M}_4)$ of the manifold. The path integral is performed over twisted $\mathbb{Z}_N$ gauge fields as in the symmetry (E.10), which is an extension of the bosonic Lorentz group $SO(4)$ by the $\mathbb{Z}_N$ gauge group. The twisted $\mathbb{Z}_N$ gauge field is a one-cochain $a$ valued in $\mathbb{Z}_N$ that satisfies

$$\delta a = (N/2)w_2(\mathcal{M}_4) \bmod N . \tag{E.11}$$

The path integral sums over all possible $a$ with fixed $w_2(\mathcal{M}_4)$ of the manifold.

If $N/2$ is odd, $\mathbb{Z}_N \cong \mathbb{Z}_{N/2} \times \mathbb{Z}_2$ and the symmetry (E.10) is isomorphic to $\mathbb{Z}_{N/2} \times Spin(4)$. Another way to see this is that (E.11) implies $w_2(\mathcal{M}_4) = \delta a \bmod 2$ by reducing both sides to mod 2. On a general manifold $w_2(\mathcal{M}_4)$ is non-trivial, and therefore the gauge field $a$ cannot be defined everywhere. Indeed, near a surface operator insertion $\oint \mathcal{B}$ that generates the one-form symmetry, the gauge field $a$ is not well-defined: a Wilson line of $a$ that links with the surface transforms by its one-form charge. For a similar discussion, see [47].

Let us return to generic $p$. On a spin manifold the theory is the same (up to a geometric counterterm) as a $\mathbb{Z}_L$ gauge theory [1]. On a non-spin manifold the situation is more interesting. For odd $N$ the equivalence to a $\mathbb{Z}_L$ gauge theory is still true [1]. However, for even $N$ a new subtlety occurs, which is related to (E.9). The computation in [1] can be interpreted to mean that when both $K = N/L$ and $p/L$ are odd (which can happen only when both $N$, $p$, and therefore also $L$ are even), or equivalently, when $pN/L^2$ is odd the equivalent $\mathbb{Z}_L$ gauge theory is actually a twisted theory as mentioned above. In terms of a $\mathbb{Z}_L$ two-form gauge field, its action is

$$\pi \frac{pN}{L^2} \int w_2(\mathcal{M}_4) \cup \mathcal{B}^{(L)} . \tag{E.12}$$

Similarly, the basic line operator in the $\mathbb{Z}_L$ gauge theory corresponding to $\exp(i \oint KA)$ also represents a fermion when $pN/L^2$ is odd.

This discussion of odd $pN/L^2$ is consistent with our 3d analysis in Section 2.4, where we saw that in this case the generating line of the $\mathbb{Z}_L$ one-form symmetry is a fermion and the 3d theory has a mixed anomaly between the $\mathbb{Z}_L$ global symmetry and gravity (E.12).

Next, consider the $\mathbb{Z}_N$ two-form gauge theory on a manifold with a boundary [1, 4].[31] We choose the Dirichlet boundary condition $B| = 0$. This explicitly breaks the one-form gauge symmetry on the boundary so the line $\widehat{V} = \exp(i \oint A)$ is liberated there and it satisfies

$$\langle \widehat{V}(\gamma)\widehat{V}(\gamma') \rangle = \frac{1}{Z} \int DADB \exp\left( i \int \frac{pN}{4\pi}BB + \frac{N}{2\pi}BdA \right) \exp\left( i \oint_\gamma A + i \oint_{\gamma'} A \right)$$
$$= \exp\left( \frac{2\pi ip}{N}\ell(\gamma, \gamma') \right), \tag{E.13}$$

---

[31]Some examples were considered in [19, 46] in the context of the Walker-Wang lattice model.

where $\gamma, \gamma' \in \partial \mathcal{M}_4$ and $\ell(\gamma, \gamma')$ is the linking number of $\gamma$ and $\gamma'$. When $L = \gcd(N, p) = 1$, the bulk theory is trivial and the $N$ lines generated by $\widehat{V}$ form the minimal Abelian TQFT $\mathcal{A}^{N,-p}$ that has a $\mathbb{Z}_N$ one-form symmetry of label $p$. For general $L$, $V = \widehat{V}^K$ can smoothly move into the bulk so it has trivial braiding. Therefore the lines on the boundary do not form a modular TQFT. However, we can perform a quotient with the bulk lines generated by $V$ to find an effective 3d TQFT $\mathcal{A}^{K,-p/L}$. If $K, p/L$ are odd, the line $V$ has half-integer spin so from the boundary perspective, $V$ can only be taken as $\psi$ the transparent spin-half line and the $2K$ lines generated by $\widehat{V}$ form a consistent spin TQFT $\mathcal{A}^{K,-p/L}$.

# F  Minimal TQFTs for general one-form symmetries

In this Appendix, we generalized the previous discussion to a general discrete one-form symmetry $\mathcal{A} = \prod \mathbb{Z}_{N_I}$.

We start with an arbitrary TQFT with one-form global symmetry $\prod \mathbb{Z}_{N_I}$ and analyze its symmetry lines, as in the introduction and in Section 2.1. Each $\mathbb{Z}_{N_I}$ factor is generated by a line $a_I$. The symmetry group means that they satisfy the mutual braiding

$$a_I^{s_I}(\gamma) a_J^{s_J}(\gamma') = a_J^{s_J}(\gamma') a_I^{s_I}(\gamma) e^{-\frac{2\pi i s_I s_J m_{IJ}}{N_I}} , \tag{F.1}$$

where $\gamma$ circles around $\gamma'$ as in Figure 2 and $m_{IJ} \in \mathbb{Z}_{N_I}$. Consistency of the mutual braiding implies $m_{IJ} N_J = m_{JI} N_I \mod N_I N_J$ and thus

$$m_{IJ} = \frac{N_I P_{IJ}}{N_{IJ}} , \qquad \text{with } N_{IJ} \equiv \gcd(N_I, N_J) , \; P_{IJ} = P_{JI} \in \mathbb{Z} . \tag{F.2}$$

This means that the spins of the symmetry lines are

$$h\left(\prod_I a_I^{s_I}\right) = \sum_{I,J} \frac{p_{IJ} s_I s_J}{2 N_{IJ}} \mod 1 , \qquad p_{IJ} = P_{IJ} \text{ or } P_{IJ} + N_{IJ} . \tag{F.3}$$

The one-form symmetry $\mathcal{A} = \prod \mathbb{Z}_{N_I}$ is characterized by the symmetric integral matrix $p_{IJ}$ that satisfies

$$p_{II} \sim p_{II} + 2N_I \quad \text{and} \quad p_{IJ} \sim p_{IJ} + N_{IJ} \text{ for } I \neq J . \tag{F.4}$$

Imposing the condition $a_I^{N_I} = 1$ requires $p_{II} N_I \in 2\mathbb{Z}$. Otherwise, the theory is a spin theory.

The braiding between $V = \prod a_I^{s_I}$ and $V' = \prod a_I^{s_I'}$ is given by

$$e^{2\pi i (h[V] + h[V'] - h[VV'])} = \exp\left(-2\pi i \sum_{I,J} \frac{p_{IJ}}{N_{IJ}} s_I s_J'\right) . \tag{F.5}$$

It will be convenient to view the braiding as a bilinear map $\mathcal{A} \times \mathcal{A} \to U(1)$. Equivalently, it defines a linear map $M : \mathcal{A} \to \widehat{\mathcal{A}} = \text{Hom}(\mathcal{A}, U(1))$.

An example of a TQFT that has the one-form symmetry $\mathcal{A} = \prod \mathbb{Z}_{N_I}$ characterized by $p_{IJ}$ is the Abelian Chern-Simons theory

$$-\sum_{I,J} \frac{p_{IJ} N_I N_J}{4\pi N_{IJ}} x^I dx^J + \sum_I \frac{N_I}{2\pi} x^I dy^I , \tag{F.6}$$

where the generating lines $a_I$ are

$$a_I = \exp\left(i \oint y^I\right) . \tag{F.7}$$

The symmetry lines in $\mathcal{L} = \ker M$ have trivial braiding with all the symmetry lines in $\mathcal{A}$. Thus the braiding (F.5) is degenerate if and only if $\mathcal{L}$ is non-trivial. If $\mathcal{L}$ is trivial, the symmetry lines form a modular 3$d$ TQFT, and we will call it the minimal Abelian TQFT for the one-form symmetry $\mathcal{A}$, denoted by $\mathcal{A}^{\{N_I\},\{p_{IJ}\}}$. An example is the $(\mathcal{Z}_N)_0$ theory that corresponds to the minimal theory with $N_1 = N_2 = N$, $p_{11} = p_{22} = 0$ and $p_{12} = p_{21} = 1$.

Next we discuss the anomaly for the one-form symmetry $\mathcal{A}$. From an argument similar to that in Section 2.4, the anomaly is characterized by the symmetric matrix $p_{IJ}$, and can be described by the following 4d term with background two-form gauge fields $\mathcal{B}_\mathcal{C} \in H^2(\mathcal{M}_4, \mathcal{A})$:

$$2\pi \int \mathcal{P}_h(\mathcal{B}_\mathcal{C}) = 2\pi \sum_I \frac{p_{II}}{2N_I} \int_{\mathcal{M}_4} \mathcal{P}(\mathcal{B}_\mathcal{C}^I) + \sum_{I<J} 2\pi \frac{p_{IJ}}{N_{IJ}} \int_{\mathcal{M}_4} \mathcal{B}_\mathcal{C}^I \cup \mathcal{B}_\mathcal{C}^J\,, \tag{F.8}$$

where on the left hand side $\mathcal{P}_h$ is the generalized Pontryagin square with the quadratic function $h$ that maps a line in $\mathcal{A}$ to its spin (F.3) (for a review see *e.g.* [39]). On the right hand side we express the anomaly in the basis $\{a_I\}$ for $\mathcal{A}$, and $\mathcal{B}_\mathcal{C}^I \in H^2(\mathcal{M}_4, \mathbb{Z}_{N_I})$ are the components of $\mathcal{B}_\mathcal{C}$ in this basis.

Let us use the anomaly (F.8) as the bulk action and promote the gauge field $\mathcal{B}_\mathcal{C}$ to be a dynamical gauge field $\mathcal{B}$. The theory has surfaces given by the fluxes of $\mathcal{B}$, and magnetic lines, both are described by the group $\mathcal{A}$ with the group multiplication given by the fusion of operators. As we will see, some of the operators have trivial correlation functions, and they should not be included in the list of non-trivial operators. The equation of motion for the gauge field $\mathcal{B}$ in (F.8) implies

$$\exp\left(2\pi i \oint M(\mathcal{B})\right) = 1\,, \tag{F.9}$$

and thus the surfaces generated by (F.9) have trivial correlation functions, while the non-trivial surfaces are described by the group $\mathcal{L} \cong \ker M$. The surfaces generated by (F.9) are described by the group $\mathcal{K} \cong \operatorname{im} M \cong \mathcal{A}/\mathcal{L}$, and the open version of them describe the line operators that have trivial correlation functions. Thus the non-trivial line operators are described by the quotient $\mathcal{L}$. The lines realize a faithful one-form symmetry $\mathcal{L}$ generated by the non-trivial surfaces. The theory can describe the spontaneous breaking of the one-form symmetry $\mathcal{A}$ generated by the surfaces to the subgroup $\mathcal{K}$ generated by the surfaces in (F.9).

Note that these $\mathcal{K}$ and $\mathcal{L}$ generalize the groups $\mathbb{Z}_K$ and $\mathbb{Z}_L$ in the case $\mathcal{A} = \mathbb{Z}_N$ that we have been discussing throughout most of this paper.

We can also study the bulk theory in the continuum description.

$$\int_{\mathcal{M}_4} \sum_{I,J} \frac{p_{IJ} N_I N_J}{4\pi N_{IJ}} B_I B_J + \sum_I \frac{N_I}{2\pi} B_I dA_I\,, \tag{F.10}$$

in terms of $U(1)$ two-form gauge fields $B_I$ and $U(1)$ one-form gauge fields $A_I$. It has a one-form gauge symmetry

$$B_I \to B_I - d\lambda_I\,, \quad A_I \to A_I + \sum_J \frac{p_{IJ} N_J}{N_{IJ}} \lambda_J\,. \tag{F.11}$$

Therefore the lines are attached to surfaces

$$\exp\left(i \oint_\gamma \sum_I s_I A_I + i \int_\Sigma \sum_I s_I \frac{p_{IJ} N_J}{N_{IJ}} B_J\right)\,, \quad \gamma = \partial\Sigma. \tag{F.12}$$

They are genuine lines, if and only if $s_I$ is in $\mathcal{L}$, the kernel of $M$. Effectively, the theory becomes a one-form (ordinary) $\mathcal{L}$ gauge theory. It may couple to $w_2(\mathcal{M}_4)$ of the manifold such that the symmetry group is twisted as described in Appendix E.

On an open manifold with the choice of boundary condition $B_I| = 0$, the gauge symmetry (F.11) is completely broken on the boundary and all the bulk lines are liberated there. Their braiding is the same as (F.5) with $p_{IJ} \to -p_{IJ}$ (see Appendix E for a similar calculation). If $\mathcal{L}$ is trivial, they form a modular TQFT $\mathcal{A}^{\{N_I\},\{-p_{IJ}\}}$. Otherwise, the bulk lines associated to $\mathcal{L}$ have trivial braiding and we can only find an effective boundary theory consisting of the lines in $\mathcal{A}/\mathcal{L}$ by modding out by the bulk lines.

Alternatively, as in the main text we can consider the boundary condition $B_I| \neq 0$. To do this, we start with a 4d-3d system with an SPT phase (F.8) in the bulk and a 3d TQFT $\mathcal{T}$ on the boundary that has an anomalous one-form symmetry coupled to the classical gauge fields $(B_{\mathcal{C}})^I$, and the anomaly is cancelled by the inflow. We can then promote the gauge fields to be dynamical. When $\mathcal{L}$ is trivial, the bulk dynamics is trivial and there is a meaningful boundary theory

$$\mathcal{T}' = \frac{\mathcal{T} \otimes \mathcal{A}^{\{N_I\},\{-p_{IJ}\}}}{\prod \mathbb{Z}_{N_I}} \,, \tag{F.13}$$

It is obtained from $\mathcal{T}$ by removing all lines that are not invariant under the one-form symmetry. When $\mathcal{L}$ is non-trivial, the theory above is not modular, and we can find an effective boundary theory as a quotient by the transparent bulk lines associated to $\mathcal{L}$. The discussion can be generalized easily to interfaces.

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
