# Peer review of "Comments on One-Form Global Symmetries and Their Gauging in 3d and 4d"

_SciPost Physics, doi:SciPost Phys. 6, 039 (2019)_

## Round 1 · Referee Report · Anonymous · 2019-2-25

Strengths

1- Physics of 3d Abelian TQFT is studied in detail with clear explanation.
2- Important factorization theorem is shown for 3d Abelian TQFTs if they have the most 't Hooft anomalous $\mathbb{Z}_N$ one-form symmetry. In this process, they naturally explain the notion of minimal TQFT.
3- They introduce the "gauging" procedure for 't Hooft anomalous $\mathbb{Z}_N$ one-form symmetry of 3d TQFT using the minimal TQFTs.
4- These results are applied to conjecture the reasonable dynamics of 4d SU(N) and PSU(N) gauge theories with $\theta$ angles. Especially, it allows us to study interfaces of PSU(N) Yang-Mills with different discrete theta parameters.
5- The paper is self contained. The necessary information to read this paper is mostly written either in the main body or in the Appendix of this paper.

Weaknesses

1- I do not come up with the weak point.

Report

The motivation of this paper is to study the interface of 4d SU(N) and PSU(N) Yang-Mills theory with different discrete theta parameters. For SU(N) Yang-Mills theory at $\theta=\pi$, this is studied in detail in Ref.[2], and the authors made the statements more precise by introducing the notion of minimal TQFT in this paper. For PSU(N) Yang-Mills theory, the 4d bulk can be a nontrivial intrinsic topological order, and the situation becomes much more complicated. The authors tackle this problem by studying the 3d Abelian topological order in this paper.

The paper is well written. Physics of 3d Abelian TQFT is explained in an explicit physical language, so the readers are not required to be too familiar with mathematics, and the necessary backgrounds, like higher-form symmetries, Abelian anyson, etc. are summarized in the Appendix. Through this paper, the authors have clarified the rigorous fact about 4d Yang-Mills theory that can be said only by symmetry and 't Hooft anomaly, and this is a useful development of our understanding of nonperturbative QFT.

From these reasons, I suggest the publication of this paper.

Requested changes

1- No changes are needed.

---

## Editorial Decision

published